# Exploring the Synthetic Potential of γ-Lactam Derivatives Obtained from a Multicomponent Reaction—Applications as Antiproliferative Agents

**DOI:** 10.3390/molecules27113624

**Published:** 2022-06-05

**Authors:** Adrián López-Francés, Xabier del Corte, Zuriñe Serna-Burgos, Edorta Martínez de Marigorta, Francisco Palacios, Javier Vicario

**Affiliations:** Departamento de Química Orgánica I, Centro de Investigación y Estudios Avanzados “Lucio Lascaray”, Facultad de Farmacia, University of the Basque Country, UPV/EHU Paseo de la Universidad 7, 01006 Vitoria-Gasteiz, Spain; adrian.lopez@ehu.eus (A.L.-F.); xabier.delcorte@ehu.es (X.d.C.); zserna001@ikasle.ehu.eus (Z.S.-B.); edorta.martinezdemarigorta@ehu.eus (E.M.d.M.)

**Keywords:** multicomponent synthesis, γ-lactams, regioselective functionalization, antiproliferative effect

## Abstract

A study on the reactivity of 3-amino α,β-unsaturated γ-lactam derivatives obtained from a multicomponent reaction is presented. Key features of the substrates are the presence of an endocyclic α,β-unsaturated amide moiety and an enamine functionality. Following different synthetic protocols, the functionalization at three different positions of the lactam core is achieved. In the presence of a soft base, under thermodynamic conditions, the functionalization at C-4 takes place where the substrates behave as enamines, while the use of a strong base, under kinetic conditions, leads to the formation of C-5-functionalized γ-lactams, in the presence of ethyl glyoxalate, through a highly diastereoselective vinylogous aldol reaction. Moreover, the nucleophilic addition of organometallic species allows the functionalization at C-3, through the imine tautomer, affording γ-lactams bearing tetrasubstituted stereocenters, where the substrates act as imine electrophiles. Taking into account the advantage of the presence of a chiral stereocenter in C-5 substituted γ-lactams, further diastereoselective transformations are also explored, leading to novel bicyclic substrates holding a fused γ and δ-lactam skeleton. Remarkably, an example of a highly stereoselective formal [3+3] cycloaddition reaction of chiral γ-lactam substrates is reported for the synthesis of 1,4-dihidropyridines, where a non-covalent attractive interaction of a carbonyl group with an electron-deficient arene seems to drive the stereoselectivity of the reaction to the exclusive formation of the *cis* isomer. In order to unambiguously determine the substitution pattern resulting from the diverse reactions, an extensive characterization of the substrates is detailed through 2D NMR and/or X-ray experiments. Likewise, applications of the substrates as antiproliferative agents against lung and ovarian cancer cells are also described.

## 1. Introduction

Among heterocyclic compounds, the γ-lactam ring is a privileged scaffold that appears as a relevant construction motif for the development of new drugs. A wide range of bioactive natural products or clinically used drugs and pharmaceuticals embody the γ-lactam scaffold as a part of their complex molecular architecture [1,2]. Thus, tremendous efforts have been devoted during the last decades towards the efficient synthesis of functionalized γ-lactam derivatives [3,4,5,6,7]. In particular, 3-pyrrolin-2-ones (also named as 1,5-dihydropyrrol-2-ones), closely related to γ-lactams, are not only useful building blocks in chemical synthesis [8,9], but also core structures of bioactive natural products and pharmaceuticals [10,11,12,13,14,15,16].

A straightforward method for the synthesis of 3-pyrrolin-2-ones consists of a three-component reaction where either pyruvate [17,18,19,20] or acetylene carboxylate [21,22,23,24,25] derivatives are mixed with aldehydes and amines, in the presence of a Brönsted acid catalyst, leading to enol-containing *iso*-tetronic acid structures or their enamine substrates, 3-amino 3-pyrrolin-2-ones. Those multicomponent protocols [26,27] are considered to be essential synthetic tools in diversity-oriented synthesis [28,29]. 3-amino 3-pyrrolin-2-ones are indeed cyclic α-dehydro α-amino acids and such a skeleton is found widely distributed among many biologically active compounds [28,29,30,31,32,33,34]. In addition, many bioactive natural products are based in a α,β-unsaturated γ-lactam structure, such as cytotoxic polyketides Myceliothermophins E, C, and D [35], cytotoxic Pukeleumid E present in *Lyngbya majuscule* algae [36], HIV-integrase inhibitor Oteromicyn [37,38], and antibiotic Pyrrocidine A [39], both of them isolated from different fungus, and in the basic structure of 1,2-dithiole group antibiotics (Holothin, Holomycin, Thiolutin, and Aureothricin) [40] (Figure 1).

In addition, from a synthetic point of view, 3-amino 3-pyrrolin-2-ones are cyclic enamine substrates and, therefore, they are outstanding tools in chemical synthesis [41]. Interestingly, some pyrrolone-based systems have been identified as p53−MDM2 [42] and STAT3 [43] inhibitors, which result in strong antiproliferative activity, and, in addition, many other molecules containing the pyrrolone scaffold have been described as efficient antitumoral agents [44,45,46,47,48].

A few years ago, we reported a Brönsted-acid-catalyzed enantioselective synthesis of 3-amino 3-pyrrolin-2-ones through a multicomponent reaction, making use of amines, aldehydes, and pyruvate derivatives as starting materials [24]. More recently, we have extended this strategy to the preparation of the corresponding phosphorus and fluorine-containing analogs, using in this case fluorine and/or phosphonate-substituted aldehydes and pyruvates [25]. Remarkably, many of those molecules have shown strong antiproliferative activity, inhibiting the growth of several carcinoma human tumor cell lines through the activation of the apoptotic mechanism [10,11]. Key features of the structure of those substrates are the very reactive cyclic enamine moiety and the presence of a chiral stereocenter at the 5-membered ring. Taking the advantage of those two characteristics, we have used 3-amino 3-pyrrolin-2-ones as substrates in Lewis-acid-catalyzed formal [3+3] annulation reactions, for the preparation of bicyclic 1,4-dihidropyridines with a remarkably high degree of stereoselectivity [12]. Considering the easiness of the multicomponent preparation of 3-amino 3-pyrrolin-2-ones and their synthetic potential in organic synthesis, we thought that a study on the synthetic applications of such lactam substrates would be of great interest in the field. Taking into account all the considerations mentioned above, here we report the synthetic applications of enamine-derived γ-lactam derivatives, obtained from a simple multicomponent reaction, with a special focus on stereoselective transformations. Moreover, in view of the relevance of the γ-lactam core in biological systems and their demonstrated properties as antitumoral agents, a study of their antiproliferative activity against carcinomic human ovarian and alveolar basal epithelial cells is also presented.

## 2. Results and Discussion

### 2.1. Chemistry

As previously described [9,21], the multicomponent reaction between aromatic amines **1**, aldehydes **2**, and pyruvate derivatives **3**, in the presence of a catalytic amount of BINOL-derived phosphoric acid, proceeds smoothly in a few hours to yield the target 3-amino 3-pyrrolin-2-ones **4**–**5** (Figure 1).

Following this procedure, several 3-amino 3-pyrrolin-2-ones **4**,**5** were synthesized and their reactivity was explored in diverse reactions. As has been addressed above, 3-amino 3-pyrrolin-2-ones derivatives **4** and **5** contain the enamine moiety in their structure. In view of the potential nucleophilic character of those substrates, first we tested the reactivity of the enamine functionality as a nucleophile using first the most simple γ-lactam substrates derived from ethyl pyruvate and formaldehyde **4a**,**b** (Figure 2). The conjugate addition of the cyclic enamines **4** to acetylene carboxylates **6** (R^2^ = OAlk) can be performed by the generation of the aza-enolate through an initial treatment with LDA at low temperature, followed by the addition of the electrophile. The reaction proceeds with a high degree of regioselectivity at the position C-4 of the five-membered heterocycle. Following this procedure, the functionalization reaction was successfully applied to methyl, ethyl, *tert*-butyl, and 2-naphthyl carboxylate derivatives **7a**–**f** in excellent yields (Figure 2).

Under the same reaction conditions, the conjugate addition of γ-lactams **4** to acetyl acetylene (R^2^ = CH_3_) failed to give the nucleophilic addition derivatives. However, the new set of reaction conditions such as stoichiometric amounts of a Lewis acid Yb(OTf)_3_ led to the formation of functionalized lactams **7g**–**h** in excellent yields (Figure 2).

γ-Lactams **7** were fully characterized on the basis of their ^1^H and ^13^C NMR and IR spectrum and HRMS. Functionalized γ-lactam **7a** was selected in order to unambiguously establish the structure of the reaction products obtained from the conjugate addition to activated alkynes. The functionalization at C-4 is evident by the presence of a singlet at δ_H_ = 4.41 ppm in ^1^H NMR spectrum that integrates to two protons, corresponding to the methylene group at the 5-membered ring and the two doublets at δ_H_ = 7.56 and 5.63 ppm for the alkene group with a typical coupling constant for a *trans* configuration of the C=C double bond (^3^*J*_HH_ = 15.7 Hz). As expected, both protons at the alkene bond proved to be coupled together in the homonuclear correlation spectroscopy (COSY) spectrum. In addition, the^13^C NMR spectrum shows a signal at δ_C_ = 49.2 ppm which was assigned to a CH_2_ carbon by distortionless enhancement by polarization transfer (DEPT) experiments. The proton and carbon signals assigned to the methylene group show a correlation in the heteronuclear single quantum coherence spectroscopy (HSQC) spectrum. Moreover, some other characteristic features of the ^13^C NMR spectrum of **6a** are the two signals corresponding to the two enamine quaternary carbons (DEPT) at δ_C_ = 136.6 and 107.5 ppm as well as the signals corresponding to the methine carbons (DEPT) of the alkene group at δ_C_ = 135.5 and 113.2 ppm.

In congruity with the proposed structure, the heteronuclear multiple bond correlation spectroscopy (HMBC) spectrum of **6a** presents two clear correlations of the methylene protons at the heterocycle with the two enamine carbons, as well as a third one with the alkene carbon next to the γ-lactam ring at δ_C_ = 135.5 ppm. As expected, the less shielded signal of the olefinic protons at δ_H_ = 7.56 ppm, corresponding to the methine group next to the heterocycle, shows correlation with the two enamine carbons and the methylene group at the 5-membered ring, as well as with the second carbon of the alkene moiety at δ_C_ = 113.2 ppm and the carbonyl of the very conjugated ester group at δ_C_ = 165.6 ppm. The most shielded signal of the olefinic protons at δ_H_ = 5.63 ppm, assigned to the CH group next to the carboxyl group, shows a relationship with the carboxyl group and the other carbon of the alkene moiety at δ_C_ = 135.5 ppm. Moreover, this olefinic proton only shows correlation with one of the quaternary enamine carbons at δ_C_ = 107.5 ppm.

In order to extend further the synthetic applications of γ-lactam substrates, next we explored the reactivity of the different functionalized derivatives **7**. Our first attempt to promote an intramolecular cyclization from acrylate substituted γ-lactam **7a** under basic conditions failed, which was attributed to an unfavorable *trans* configuration of the alkene bond. For this reason, next, we performed the double reduction of the alkene and enamine moieties under hydrogen pressure in the presence of palladium as the catalyst, affording saturated γ-lactam derivative **8** in excellent yield with a complete degree of *syn*-diastereoselection (Figure 3). As a consequence of the less strained structure obtained in the saturated derivative **8**, in this case the treatment under basic conditions leads to the formation of bicyclic γ-lactam **9** in almost quantitative yield (Figure 3). As far as we are concerned, this represents the first example of a synthetic methodology leading to the diastereoselective formation of a bicyclic substrate comprising both γ and δ-lactam units. Additionally, the selective hydrolysis of the enamine moiety can be performed under acidic conditions to yield isotetramic acid derivative **10** in an almost quantitative yield.

The relative configuration of the stereocenters in γ-lactams **8** and **9** was determined by nuclear Overhauser enhancement spectroscopy (NOESY) experiments (see ESI). NMR experiments with compound **8** showed a NOE of 3.98% between both protons at the two stereogenic centers, at δ = 4.13 and 2.91 ppm, respectively, indicating that these two atoms are oriented in the same direction. This relative configuration is in agreement with the accepted mechanism for catalytic hydrogenation of a carbon-carbon double bond, consisting in a *syn* addition of hydrogen.

Continuing with our interest in the nucleophilic addition reactions of enamine-derived γ-lactams, next, we explored the functionalization of substrates **4c**,**d**, holding a phosphorated substituent at the position 4 of the ring. Accordingly, the treatment of γ-lactams **4c**,**d** with LDA at low temperature, followed by the addition of ethyl glyoxalate, led to the nucleophilic addition products **12a**,**b**, where the 5-membered ring is selectively functionalized at C-5. In this case, the presence of the bulky phosphonate or phosphine oxide substituents may block the β-enamine functionalization and the reaction presumably proceeds through the generation of aromatic enol-type intermediate **11**, under kinetic conditions instead of the aza-enolate. Then, the subsequent selective vinylogous nucleophilic addition of anionic species **11** to the aldehyde electrophile proceeds selectively at position 5 of the 5-membered ring, leading to the formation of α-hydroxyester-functionalized γ-lactams **12a**,**b** in good yields (Figure 4). It should be noted that these are the first examples of C-5-functionalized phosphorus-substituted γ-lactams holding an enamine structure, since such substrates are known to rapidly provide the enol-derived γ-lactams through the spontaneous hydrolysis of the enamine moiety, as a consequence of the high steric hindrance present in the 5-membered heterocycle [21].

Remarkably, the nucleophilic addition reaction proceeds with a high degree of stereoselectivity and a single diastereoisomer is observed in the NMR spectra of the crude reaction. Unfortunately, NOESY experiments on the functionalized γ-lactams did not provide solid evidence that could lead to the unambiguous determination of the relative configuration of the adduct. For this reason, a single crystal of substrate **12b** was isolated from a mixture of CH_2_Cl_2_/hexanes and, next, the structure obtained by X-ray diffraction experiments revealed unequivocally a relative configuration derived from the *anti*-addition of the γ-lactam core to the aldehyde electrophile (Figure 2). A plausible transition state leading to the *anti*-derivative may consist of an electrostatic interaction between the aldehyde and the lithium cation, directing the carbonyl group towards the pyrrole ring. A possible additional stabilization by hydrogen bonding between the NH substituent and the carbonyl oxygen may provide a very rigid arrangement of the aldehyde, which orients its terminal hydrogen in a *gauche* conformation relative to the bulky phosphorus substituent, as drawn in the Newman projection shown in Figure 2.

Surprisingly, under the same reaction conditions, for the special case of the nucleophilic addition of phosphorated γ-lactam **4d** to *p*-nitrophenyl isocyanate, the introduction of an imine moiety into the five-membered ring was observed to give γ-lactam **13** (Figure 5). An explanation for this behavior may consist of an initial formation of the amide **14** by the typical nucleophilic addition to an isocyanate substrate. Then, a spontaneous oxidation of the 5-membered heterocycle is proposed to yield intermediate **15**. Next, an intramolecular conjugate addition of the amide nitrogen to the heterocycle would afford the instable three-membered aziridinone species **16**, and a subsequent rearrangement of the aziridinone and the enamine moieties at the γ-lactam core would lead to *N*-formamide **17** that undergoes a spontaneous cleavage of the formyl residue to afford imine-functionalized γ-lactam **13**.

Due to the complexity of the transformation proposed and, in view of the simple pattern obtained in the ^1^H NMR spectrum, which only shows aromatic protons and the acidic NH group, we were still skeptical about the real identity of the substrate obtained in the reaction. Although the exact mass was consistent with the proposed structure, in order to wipe away any doubt regarding the identity of substrate **13**, a monocrystal of γ-lactam derivative **13** was isolated and its structure was unambiguously established by X-ray diffraction (Figure 3). Key features of the crystal structure of **13** are the parallel conformation of the *N*-aryl groups at the lactam nitrogen and the imine group, possibly stabilized by π-stacking, and the intramolecular hydrogen bonding observed between the enamine NH and the phosphoryl oxygen.

Next, the functionalization reaction was extended to the use of chiral γ-lactam derivatives **5** holding a substituent at position 5 of the heterocycle. Under similar conditions, the treatment of substituted γ-lactam **5a** with LDA at low temperature, followed by the addition of methyl iodide, afforded the alkylated substrate **18** in moderate yield (51%), together with a small amount of the hydrolyzed enol-derived lactam **19** (26%) (Figure 6). The same behavior has already been observed in similar highly functionalized 3-amino 3-pyrrolin-2-ones and was attributed to the high steric hindrance at the 5-membered ring, due to the high degree of substitution [20,21,22,23,24,25]. In addition, enol-derived substrate **19** can be prepared in an almost quantitative yield by the hydrolysis of enamine-derived γ-lactam **18** in 0.1 M aqueous hydrochloric acid. Unfortunately, under the same reaction conditions, the nucleophilic addition of γ-lactam **5a** to other electrophiles, such as acetylene carboxylates, acrylates, or bulkier alkyl halides, failed, which was attributed to the high steric crowding present in the expected densely functionalized resulting substrates.

The alkylation at C-5 in γ-lactam derivative **18** was verified by an extensive NMR study. The selective functionalization is deduced in the ^1^H NMR spectrum from the disappearance of the two doublets at δ_H_ = 6.01 and 5.63 ppm (^3^*J*_HH_ = 2.6 Hz), corresponding to the two CH groups of the γ-lactam ring in starting substrate **5a**, and the appearance of a new singlet at δ_H_ = 5.33 ppm assigned, in principle, to the proton at the chiral carbon of the heterocycle in **18**. The presence of the methyl group is evident from the presence of an intense singlet at δ_H_ = 1.60 ppm.

The most characteristic signals found in the ^13^C NMR spectrum of **18** are the four carbons of the 5-membered heterocycle, comprising the chiral CH carbon at δ_C_ = 67.6 ppm, the two quaternary enamine carbons at δ_C_ = 125.7 and 131.3 ppm, and the conjugated amide carbonyl group at δ_C_ = 168.2 ppm. Likewise, the presence of a dimethylaminomethyl group is evidenced from the appearance of an intense signal at δ_C_ = 13.6 ppm. The multiplicity of all carbon signals was verified by DEPT experiments. In addition, a strong C-H interaction was observed in HSQC spectrum between the ^1^H NMR and ^13^C NMR signals attributed to the CH, and methyl groups.

HMBC experiments provide additional evidence for the functionalization at C-4 in substrate **18**. Accordingly, a correlation is observed for both, the CH and methyl protons, with the two quaternary carbons of the enamine moiety. Moreover, the ^1^H NMR signal assigned to the methyl group shows a correspondence with the stereogenic carbon at the 5-membered heterocycle, while the methine proton presents the reciprocal correlation.

On the other hand, the presence of an ester group at the stereogenic carbon of the γ-lactam ring provides an additional acidic character to the proton at C-5. Accordingly, the vinylogous Mannich reaction of 3-pyrrolin-2-one **5g** is observed upon treatment with trimethylamine in the presence of Eschenmoser salt, providing, in good yield, γ-lactam **20**, bearing a tetrasubstituted carbon (Figure 7).

In order to provide evidence of the selective functionalization at C-5, an extensive NMR study of substrate **20** was implemented. In this case, the ^1^H NMR spectrum of substrate **20** shows a singlet at δ_H_ = 5.95 ppm for the methine group and two additional doublets at δ_H_ = 2.74 and 3.17 ppm, characteristic of the two diastereotopic protons of the methylene group, showing a typical *geminal* coupling constant (^2^*J*_HH_ = 14.0 Hz). As expected, the doublets corresponding to the diastereotopic protons show a strong interaction in the COSY spectrum.

The most characteristic signals found in the ^13^C NMR spectrum are the four carbons of the 5-membered heterocycle, comprising the chiral quaternary carbon at δ_C_ = 72.7 ppm, the two enamine carbons at δ_C_ = 105 and 134 ppm, and the amide carbonyl group at δ_C_ = 168.3 ppm. The functionalization at C-5 in **20** is deduced from the appearance of the quaternary carbon at higher field than in C-4-functionalized substrate **18** (72.7 vs. 125.7 ppm), suggesting a sp^3^ hybridization of such carbon. Moreover, the fact that the CH carbon appears at significantly lower field in the ^13^C NMR spectrum of **20** than in substrate **17** (104.9 vs. 67.6 ppm) suggests that this signal corresponds to an olefinic methine carbon. As usual, the multiplicity of the carbons was verified by DEPT experiments and could also be deduced from the interactions observed in the HSQC spectrum.

Some additional evidences of the functionalization at C-5 in substrate **20** can be gathered from HMBC experiments. In this case, the principal argument to consider a C-5 functionalization arises from the strong correlation observed between the diastereotopic methylene protons and the carboxylic carbon, corresponding to the ester moiety. Moreover, both diastereotopic protons show an additional interaction with the enamine CH group, while no correspondence with the quaternary enamine carbon is observed.

Considering the presence of a chiral stereocenter in C-5-substituted γ-lactam substrates **5**, next, some other diastereoselective transformations were explored. A simple transformation of substrate **5b** consists of the hydrogenation of the endocyclic enamine moiety under hydrogen pressure (80 psi) in the presence of palladium as catalyst, leading to the formation of a single diastereoisomer of saturated γ-lactam derivative **21** in excellent yield (Figure 8).

The relative configuration of the stereocenters in saturated γ-lactam **21** was determined by NOESY (see ESI). The fact that the signals corresponding to both protons at the two stereogenic centers, at δ = 4.78 and 4.16 ppm, show a strong NOE effect with the same proton at the diastereotopic methylene group at δ = 3.10 ppm, suggests that these three atoms are oriented in the same direction. In addition, no significant NOE is observed between the signal corresponding to the second of the diastereotopic protons, at δ = 2.08 ppm, and the two protons at both stereogenic centers, which confirms a relative *cis* configuration between the carboxylate and the amino groups in structure **21**. Thus, the conventional mechanism for a catalytic hydrogenation of a carbon-carbon double bond is deduced from this configuration, where a *syn* addition of hydrogen takes place from the less hindered face, that is, the opposite to the carboxylate substituent at the stereogenic carbon.

In addition, similar cyclic enamines have proved to be excellent substrates in [3+3] annulation reactions [12]. For this reason, enamine-derived chiral lactam **5b** was treated with β,γ-unsaturated α-ketoester **22** in the presence of a catalytic amount of ytterbium triflate, leading to the formation of a single diastereoisomer of bicyclic dihydropyridine **23** in excellent yield (Figure 8). In this case, NOESY experiments showed a strong correlation between both protons at the chiral centers at δ = 4.93 and 5.05 ppm, revealing a *cis* relative configuration of both substituents at the stereogenic carbons (see ESI). In this case, in agreement with the proposed pathway for this reaction [49], the mechanism of the process may consist of a Michael addition of the enamine moiety to the conjugated double bond of β,γ-unsaturated α-ketoester and a second nucleophilic addition of the enamine nitrogen to the carbonyl group. A non-covalent attractive interaction of the carbonyl moiety with the electron-deficient arene bearing the nitro-electron-withdrawing substituent [50] may drive the stereoselectivity of the reaction to the exclusive formation of the *cis* diastereoisomer as shown in Figure 4.

Next, taking the advantage of the possible in situ tautomerization of the enamine into an imine moiety, the potential of unsaturated γ-lactam substrates as electrophiles was studied. Accordingly, the treatment of 5-substituted γ-lactam **5a** derived from *p*-toluidine (R^1^ = *p*-MeC_6_H_4_) with methyllithium in THF at low temperature led to the efficient formation of unsaturated lactam derivative **25a**, holding a tetrasubstituted chiral carbon (Figure 9). Next, the reaction was extended to the use of *n*-butyllithium, obtaining almost identical results under the same reaction conditions (Figure 9, **25b**). The scope of the reaction using *n*-butyllithium as nucleophilic reagent was also extended to the use of other γ-lactam substrates **5c**–**e**, derived from different aromatic amines such as *p*-anisidine, *p*-bromoaniline and *m*-chloroaniline (R^1^ = *p*-MeOC_6_H_4_, *p*-BrC_6_H_4_, *m*-ClC_6_H_4_), providing tetrasubstituted γ-lactam derivatives **25c**–**e** also in very good yields (Figure 9). 

The formation of substrates **25** was evident by the presence of a chemical shift in the ^1^H NMR spectrum of compound **25a** (R^1^ = *p*-Me-C_6_H_4_, R^2^ = Me) at δ = 5.63 ppm, corresponding to the olefinic proton at the 5-membered ring, together with the two characteristic signals in ^13^C NMR spectrum at δ = 131.9 and 94.8 ppm, which correspond to the quaternary carbon and the CH group of the alkene bond, respectively. Moreover the exact mass of compound **25a** showed a value of 368.1889 amu, which is in agreement with the proposed formula. In addition, the heteronuclear multiple bond correlation experiment (HMBC) showed a correlation between the singlet in ^1^H NMR, corresponding to the methyl group at δ = 1.45 ppm, and the signals of the alkene CH and the amide carbonyl groups in ^13^C NMR at δ = 94.8 and 175.5 ppm, respectively, while no correlation was observed between the methyl and the phenyl substituents (see ESI). These observations imply unequivocal evidence of the substitution at C-3 of the γ-lactam ring.

The reaction seems to proceed through an initial nucleophilic addition of the organometallic reagent to the imine tautomer of **5**, followed by spontaneous oxidation of the 5-membered ring under exposure to the atmosphere (Figure 9). In order to prove this hypothesis, the hydrogenation reaction of the alkene bond was performed in the presence of a catalytic amount of Pd on carbon. After a quick filtration of the reaction, the ^1^H NMR spectrum of the crude reaction showed the presence of the characteristic signals corresponding to the diastereotopic protons of the CH_2_ group of saturated intermediate **24** that rapidly disappeared to show again the spectrum corresponding to **25a** (Figure 9). However, we were unable to isolate a pure sample of intermediate **24**.

### 2.2. Biological Results

As has been addressed above, some 3-pyrrolin-2-one derivatives have been demonstrated to induce apoptosis in several cancer cell lines [10,11]. For this reason and in order to underscore the usefulness of our functionalized γ-lactam substrates, next, the in vitro cytotoxicity of some of the heterocycles was evaluated by testing their antiproliferative activities against several human cancer cell lines. Cell counting kit (CCK-8) assay was used for the evaluation of growth inhibition. Moreover, non-malignant MRC5 lung fibroblasts were tested for studying selective toxicity [51]. Chemotherapeutic doxorubicin was used as reference value and 4,5-unsubstituted 3-amino 3-pyrrolin-2-one **27** [11] (R^1^ = *p*-MeC_6_H_4_, R^2^ = R^3^ = H) was used in order to evaluate the influence of the introduction of the different substituents.

First, the cytotoxicity of the simplest substrates **7** derived from C-5-unsubstituted γ-lactams was evaluated. Although most of alkene-substituted γ-lactams **7** showed no activity, substrate **7d** derived from *p*-toluidine and obtained from the functionalization with *tert*-butyl acetylene carboxylate presented some toxicity against SKOV3 cell line with an IC_50_ value of 21.21 ± 2.36 μM and high selectivity towards A549 and MRC5 cell lines (Table 1, Entry 1).

Next, the growth inhibition activity of chiral C-5-substituted γ-lactams was explored. In this context, due to their chemical similitude to natural phosphate metabolites, phosphonate derivatives show multiple biological activities and, for this reason, they have found numerous applications in medicine and agrochemistry [52,53,54,55,56]. For this reason, the antiproliferative activity of γ-lactam derivative **12a** bearing a phosphonate group at C-4 was studied. However, for this special case, no significant toxicity was observed in A549 or SKOV3 cell lines (Table 1, Entry 2). In addition, although the appearance of phosphine oxides in drug discovery is rare, if compared with their counterparts, phosphates, phosphonates, or phosphoramidates, a few of those derivatives have proven to be excellent drug candidates such as anticancer drugs ridaforolimus [57,58] or brigatinib [59,60]. Accordingly, phosphine-oxide-substituted γ-lactam derivative **12b** presented an excellent IC_50_ value of 3.6 ± 0.20 μM with also high selectivity towards SKOV3 and MRC5 cell lines (Table 1, Entry 3). The IC_50_ values observed for compound **12b** suggest that the presence of the phosphine oxide moiety is critical in order to obtain both good values of toxicity and selectivity (Table 1, Entry 3 vs. Entry 9). It should be noted that other attempts to measure the cytotoxicity of C-5-unsubstituted phosphorated γ-lactams, like **4d**, failed due to the low solubility of phosphine oxides in the solvents compatible with the biological tests [11]. However, in this case, the C-5 functionalization of **4d** with ethyl glyoxalate provides a solvent-friendly structure in **12b**, suitable for the evaluation of its antiproliferative activity.

Continuing with the SAR study on the γ-lactam derivatives, the cytotoxicity of unsaturated lactams **25** bearing a tetrasubstituted carbon was evaluated. Substrate **25a** derived from *p*-toluidine and, when holding a methyl substituent at the tetrasubstituted carbon, showed good toxicity against A549 and SKOV3 cell lines, with IC_50_ values of 15.16 ± 1.84 and 9.73 ± 1.27 μM, respectively. However, compound **25a** did not show any selectivity towards non-malignant cells (Table 1, Entry 4). Substitution of the methyl group by a more lipophilic *n*-butyl group in γ-lactam derivative **25b** resulted in an increase of the cytotoxicity, showing IC_50_ values of 3.35 ± 0.42 and 6.93 ± 0.35μM in A549 and SKOV3 cell lines, respectively. Nonetheless, a similar increase in the toxicity towards MRC5 cell line was also observed in this case (Table 1, Entry 5). Almost identical results were obtained for other *n*-butyl-substituted γ-lactams **25c**–**e** derived from different anilines with strong toxicity against cancer cells but non-selectivity towards non-malignant ones (Table 1, Entries 6–8). The best candidate of this family was found to be *p*-chloroaniline-derived lactam **25e**, which showed IC_50_ values of 4.71 ± 1.40 and 4.64 ± 0.62 μM in A549 and SKOV3 cell lines, respectively, and an IC_50_ value five times higher in MRC5 cell line (Table 1, Entry 8). The comparison of the IC_50_ values obtained for compounds **25** with the ones reported for unsubstituted lactam **25** suggests that either the presence of the tetrasubstituted carbon or the deconjugation of the C=C double bond results in highly toxic species (Table 1, Entries 4–8 vs. Entry 9).

It is also worth mentioning that, with the exception of substrate **25d**, all other γ-lactam derivatives fulfill the requirements for orally active drugs in humans, in accordance with Lipinski’s rule of five. According to the predictions, most of the described substrates have high gastrointestinal absorption and the ability to cross the blood-brain-barrier (see Table in ESI).

## 3. Materials and Methods

### 3.1. Chemistry

#### 3.1.1. General Experimental Information

Solvents for extraction and chromatography were technical grade. All solvents used in reactions were freshly distilled from appropriate drying agents before use. All other reagents were recrystallized or distilled as necessary. All reactions were performed under an atmosphere of dry nitrogen. Analytical TLC was performed with silica gel 60 F_254_ plates. Visualization was accomplished by UV light. ^1^H, ^13^C, ^31^P, and ^19^F-NMR spectra were recorded on a Varian Unity Plus (Varian Inc., NMR Systems, Palo Alto (CA), United States) (at 300 MHz, 75 MHz, 120 MHz, and 282 MHz, respectively) and on a Bruker Avance 400 (Bruker BioSpin GmbH, Rheinstetten, Germany) (at 400 MHz for ^1^H, and 100 MHz for ^13^C). Chemical shifts (δ) are reported in ppm relative to residual CHCl_3_ (δ = 7.26 ppm for ^1^H and δ = 77.16 ppm for ^13^C NMR) and using phosphoric acid (50%) or HF as external reference (δ = 0.0 ppm) for ^31^P and ^19^F NMR spectra. Coupling constants (*J*) are reported in Hertz. Data for ^1^H NMR spectra are reported as follows: chemical shift, multiplicity, coupling constant, integration. Multiplicity abbreviations are as follows: s = singlet, d = doublet, t = triplet, q = quartet, m = multiplet. ^13^C NMR peak assignments were supported by distortionless enhanced polarization transfer (DEPT). High-resolution mass spectra (HRMS) were obtained by positive-ion electrospray ionization (ESI). Data are reported in the form *m*/*z* (intensity relative to base = 100). Infrared spectra (IR) were taken in a Nicolet iS10 Thermo Scientific spectrometer (Thermo Scientific Inc., Waltham, Massachusetts (MA), United States) as neat solids. Peaks are reported in cm^−1^.

#### 3.1.2. Compounds Purity Analysis

All synthesized compounds were analyzed by HPLC to determine their purity. The analyses were performed on Agilent 1260 infinity HPLC system (Agilent, Santa Clara, CA, United States) using a CHIRALPAK^®^ IA column (5 μm, 0.54 cm ø × 25 cm, Daicel Chiral Technologies, Illkirch Cedex, France) at room temperature. All the tested compounds were dissolved in dichloromethane, and 5 μL of the sample was loaded onto the column. Ethanol and heptane were used as the mobile phase, and the flow rate was set at 1.0 mL/min. The maximal absorbance at the range of 190–400 nm was used as the detection wavelength. The purity of all the derivatives tested in biological essays is >95%, which meets the purity requirement by the Journal.

#### 3.1.3. Experimental Procedures and Characterization Data for Compounds **13**, **15**, **16**, and **23**

##### General Procedure for the Multicomponent Synthesis of γ-Lactams **4** and **5**

Following a known procedure [11], a solution of amine **1** (10 mmol), aldehyde **2** (5 mmol), ethyl pyruvate **3** (1.74 g, 15 mmol), and 1,1′-binaphthyl-2,2′-diyl hydrogen phosphate (174 mg, 10 mol%) was stirred in Et_2_O (25 mL) at room temperature or MTBE (25 mL) at 55 °C (heating plate with Heat-On) in the presence of anhydrous MgSO_4_ (2.5 g) for 48 h. The volatiles were distilled off at reduced pressure and the crude residue was purified by chromatography (AcOEt/hexanes 9:1) to afford pure γ-lactams **4** and **5**.

##### General Procedure for the Nucleophilic Conjugate Addition of γ-Lactams **4** to Activated Alkynes

Procedure A (Esters). The corresponding γ-lactam **4** (1 mmol) was added to a freshly prepared solution of LDA (1.2 mmol) in THF (3 mL) at −78 °C under N_2_ atmosphere. After 1 h, the selected alkyl propiolate (1.2 mmol) was added and the reaction was stirred at room temperature until the disappearance of the starting materials (monitored by TLC and/or ^1^H RMN, ~4 h). The reaction was acidified with a 0.5 M aqueous solution of HCl (10 mL), and extracted with dichloromethane (2 × 10 mL). The combined organic layers were washed with water (2 × 10 mL), dried over MgSO_4_, and concentrated under vacuum. The crude residue was purified by crystallization, affording pure functionalized γ-lactam derivatives **6**. In some cases, a previous purification by chromatography was necessary as detailed for each compound.

Procedure B (Ketones). A solution of the corresponding γ-lactam **4** (1 mmol), 3-butn-2-one (0.097 mL, 1.2 mmol) and Yb(OTf)_3_ (1 mmol, 620 mg) in dichloromethane (3 mL) was stirred at room temperature until the disappearance of the starting (monitored by TLC, 4 to 20 h). The reaction mixture was filtered through celite and concentrated under vacuum and the crude residue was purified by crystallization, affording pure functionalized γ-lactam derivatives **7**. In some cases, previous purification by chromatography column was necessary as detailed for each compound.

Methyl (E)-3-(5-oxo-1-(*p*-tolyl)-4-(*p*-tolylamino)-2,5-dihydro-1*H*-pyrrol-3-yl)acrylate (**7a**). The general procedure A was followed using 1-(*p*-tolyl)-3-(*p*-tolylamino)-1,5-dihydro-2*H*-pyrrol-2-one **4a** (278 mg, 1 mmol) and methyl propiolate (107 μL, 1.2 mmol), affording 304 mg (84%) of **7a** as yellow crystals after crystallization (Dichloromethane/Hexanes 1:3). M.p. (Dichloromethane/Hexanes) = 189–191 °C. ^1^H NMR (400 MHz, CDCl_3_) δ 7.66 (d, ^3^*J*_HH_ = 8.6 Hz, 2H, 2 × CH_Ar_), 7.56 (d, ^3^*J*_HH_ = 15.7 Hz, 1H, CH=), 7.22 (d, ^3^*J*_HH_ = 8.6 Hz, 2H, 2 × CH_Ar_), 7.19 (d, ^3^*J*_HH_ = 8.4 Hz, 2H, 2 × CH_Ar_), 7.11 (d, ^3^*J*_HH_ = 8.4 Hz, 2H, 2 × CH_Ar_), 6.74 (s, 1H, NH), 5.63 (d, ^3^*J*_HH_ = 15.7 Hz, 1H, CH=), 4.41 (s, 2H, CH_2_), 3.68 (s, 3H, OCH_3_), 2.37 (s, 3H, CH_3_Tol), 2.36 (s, 3H, CH_3_Tol) ppm. ^13^C NMR {^1^H} (101 MHz, CDCl_3_) δ 167.4 (C=O), 165.6 (C=O), 136.6 (C_quat_), 136.4 (C_quat_), 136.3 (C_quat_), 135.5 (CH=), 135.1 (C_quat_), 134.8 (C_quat_), 130.2 (2 × CH_Ar_), 129.9 (2 × CH_Ar_), 122.5 (2 × CH_Ar_), 119.1 (2 × CH_Ar_), 113.2 (CH=), 107.5 (C_quat_), 51.6 (OCH_3_), 49.2 (CH_2_), 21.1 (CH_3_Tol), 21.0 (CH_3_Tol) ppm. FTIR (neat) ν_max_: 3321 (NH _st_), 3059 (=CH _st_), 1701 (C=O _st_), 1676 (C=O _st_), 1611 (C=C _st_), cm^−1^. HRMS (ESI-TOF) *m/z*: [M+H]^+^ calcd for C_22_H_23_N_2_O_3_ 363.1709, Found 363.1708. 

Methyl (*E*)-3-(5-oxo-1-(3-(trifluoromethyl)phenyl)-4-((3-(trifluoromethyl)phenyl)amino)- 2,5-dihydro-1*H*-pyrrol-3-yl)acrylate (**7b**). The general procedure A was followed using 1-(3-(trifluoromethyl)phenyl)-3-((3-(trifluoromethyl)phenyl)amino)-1,5-dihydro-2*H*-pyrrol-2-one **4b** (386 mg, 1 mmol) and methyl propiolate 107 μL, 1.2 mmol), affording 285 mg (61%) of **7b** as a red oil after chromatography (Hexane/AcOEt 9:1). ^1^H NMR (400 MHz, CDCl_3_) δ 8.10–7.99 (m, 2H, 2 × CH_Ar_), 7.59–7.44 (m, 3H, 3 × CH_Ar_), 7.54 (d, ^3^*J*_HH_ = 15.8 Hz, 1H, CH=), 7.40 (d, ^3^*J*_HH_ = 7.4 Hz, 2H, 2 × CH_Ar_), 7.33 (d, ^3^*J*_HH_ = 7.4 Hz, 1H, CH_Ar_), 6.82 (s, 1H, NH), 5.80 (d, ^3^*J*_HH_ = 15.8 Hz, 1H, CH=), 4.54 (s, 2H, CH_2_), 3.71 (s, 3H, CH_3_) ppm. ^13^C NMR {^1^H} (101 MHz, Acetone *d*_6_) δ 167.0 (C=O), 166.6 (C=O), 143.0 (C_quat_), 141.0 (C_quat_), 136.0 (C_quat_), 134.6 (CH=), 131.6 (q, ^2^*J*_CF_ = 32.3 Hz, C_quat_), 131.6 (q, ^2^*J*_CF_ = 31.9 Hz, C_quat_), 130.9 (2 × CH_ar_), 125.2 (q, ^1^*J*_CF_ = 271.6 Hz, CF_3_), 125.1 (q, ^1^*J*_CF_ = 271.3 Hz, CF_3_), 124.2 (CH_ar_), 122.4 (CH_ar_), 121.2 (q, ^3^*J*_FC_ = 4.0 Hz, CH_Ar_), 119.8 (q, ^3^*J*_FC_ = 3.8 Hz, CH_Ar_), 117.9 (CH=), 117.7 (C_quat_), 117.2 (q, ^3^*J*_FC_ = 3.9 Hz, CH_Ar_), 115.6 (q, ^3^*J*_FC_ = 4.1 Hz, CH_Ar_), 51.7 (CH_3_), 49.6 (CH_2_) ppm. ^19^F-NMR (282 MHz, CDCl_3_) δ −63.1, −63.5 ppm. FTIR (neat) ν_max_: 3309 (NH _st_), 3062 (=CH _st_), 1706 (C=O _st_), 1679 (C=O _st_), 1609 (C=C _st_) cm^−1^. HRMS (ESI-TOF) *m/z*: [M+H]^+^ calcd for C_22_H_17_F_6_N_2_O_3_ 471.1143, Found 471.1140. 

Ethyl (*E*)-3-(5-oxo-1-(*p*-tolyl)-4-(*p*-tolylamino)-2,5-dihydro-1*H*-pyrrol-3-yl)acrylate (**7c**). The general procedure A was followed using 1-(*p*-tolyl)-3-(*p*-tolylamino)-1,5-dihydro-2*H*-pyrrol-2-one **4a** (278 mg, 1 mmol) and ethyl propiolate (118 μL, 1.2 mmol), affording 345 mg (91%) of **7c** as brown crystals after crystallization (Dichloromethane/Hexanes 1:3). M.p. (Dichloromethane/Hexanes) = 211–212 °C. ^1^H NMR (400 MHz, CDCl_3_) δ 7.65 (d, ^3^*J*_HH_ = 8.4 Hz, 2H, 2 × CH_Ar_), 7.54 (d, ^3^*J*_HH_ = 15.7 Hz, 1H, CH=), 7.22 (d, ^3^*J*_HH_ = 8.4 Hz, 2H, 2 × CH_Ar_), 7.19 (d, ^3^*J*_HH_ = 8.3 Hz, 2H, 2 × CH_Ar_), 7.11 (d, ^3^*J*_HH_ = 8.3 Hz, 2H, 2 × CH_Ar_), 6.75 (s, 1H, NH), 5.62 (d, ^3^*J*_HH_ = 15.7 Hz, 1H, CH=), 4.40 (s, 2H, CH_2_), 4.12 (q, ^3^*J*_HH_ = 7.1 Hz, 2H, CH_2_CH_3_), 2.35 (s, 6H, 2 × CH_3_Tol), 1.23 (t, ^3^*J*_HH_ = 7.1 Hz, 3H, CH_2_CH_3_) ppm. ^13^C NMR {^1^H} (101 MHz, CDCl_3_) δ 166.9 (C=O), 165.6 (C=O), 136.6 (C_quat_), 136.3 (C_quat_), 136.3 (C_quat_), 135.2 (C_quat_), 135.2 (C_quat_), 134.8 (CH=), 130.2 (2 × CH_Ar_), 129.9 (2 × CH_Ar_), 122.8 (2 × CH_Ar_), 119.0 (2 × CH_Ar_), 113.6 (CH=), 107.4 (C_quat_), 60.3 (CH_2_CH_3_), 49.2 (CH_2_), 21.1 (CH_3_Tol), 21.0 (CH_3_Tol), 14.3 (CH_2_CH_3_) ppm. FTIR (neat) ν_max_: 3381 (NH _st_), 3066 (=CH _st_), 1698 (C=O _st_), 1676 (C=O _st_), 1605 (C=C _st_) cm^−1^. HRMS (ESI-TOF) *m*/*z*: [M+H]^+^ calcd for C_23_H_25_N_2_O_3_ 377.1865, Found 377.1863. 

Tert-butyl (*E*)-3-(5-oxo-1-(*p*-tolyl)-4-(*p*-tolylamino)-2,5-dihydro-1*H*-pyrrol-3-yl)acrylate (**7d**). The general procedure A was followed using 1-(*p*-tolyl)-3-(*p*-tolylamino)-1,5-dihydro-2*H*-pyrrol-2-one **4a** (278 mg, 1 mmol) and *tert*-butyl propiolate (166 μL, 1.2 mmol), affording 388 mg (96%) of **7d** as orange crystals after crystallization (Dichloromethane/Hexanes 1:3). M.p. (Dichloromethane/Hexanes) = 186–187 °C. ^1^H NMR (400 MHz, CDCl_3_) δ 7.66 (d, ^3^*J*_HH_ = 8.6 Hz, 2H, 2 × CH_Ar_), 7.44 (d, ^3^*J*_HH_ = 15.7 Hz, 1H, CH=), 7.22 (d, ^3^*J*_HH_ = 8.6 Hz, 2H, 2 × CH_Ar_), 7.18 (d, ^3^*J*_HH_ = 8.3 Hz, 2H, 2 × CH_Ar_), 7.11 (d, ^3^*J*_HH_ = 8.3 Hz, 2H, 2 × CH_Ar_), 6.68 (s, 1H, NH), 5.55 (d, ^3^*J*_HH_ = 15.7 Hz, 1H, CH=), 4.40 (s, 2H, CH_2_), 2.36 (s, 3H, CH_3_Tol), 2.34 (s, 3H, CH_3_Tol), 1.41 (s, 9H, 3 × CH_3_*t*Bu) ppm. ^13^C NMR {^1^H} (101 MHz, CDCl_3_) δ 166.1 (C=O), 165.8 (C=O), 136.7 (C_quat_), 136.4 (C_quat_), 135.8 (C_quat_), 134.9 (C_quat_), 134.7 (C_quat_), 134.2 (CH=), 130.1 (2 × CH_Ar_), 129.9 (2 × CH_Ar_), 122.8 (2 × CH_Ar_), 119.0 (2 × CH_Ar_), 116.0 (CH=), 107.7 (C_quat_), 80.1 (C_quat_*t*Bu), 49.2 (CH_2_), 28.24 (3 × CH_3_*t*Bu), 21.1 (CH_3_Tol), 21.0 (CH_3_Tol) ppm. FTIR (neat) ν_max_: 3402 (NH _st_), 3041 (=CH _st_), 1694 (C=O _st_), 1679 (C=O _st_), 1615 (C=C _st_) cm^−1^. HRMS (ESI-TOF) *m/z*: [M+H]^+^ calcd for C_25_H_29_N_2_O_3_ 405.2178, Found 405.2179. 

*Tert*-butyl (*E*)-3-(5-oxo-1-(3-(trifluoromethyl)phenyl)-4-((3-(trifluoromethyl)phenyl) amino)-2,5-dihydro-1*H*-pyrrol-3-yl)acrylate (**7e**). The general procedure A was followed using 1-(3-(trifluoromethyl)phenyl)-3-((3-(trifluoromethyl)phenyl)amino)-1,5-dihydro- 2*H*-pyrrol-2-one **4b** (386 mg, 1 mmol) and *tert*-butyl propiolate (166 μL, 1.2 mmol), affording 380 mg (74%) of **7e** as a yellow oil after chromatography (Hexanes/AcOEt 9:1). ^1^H NMR (400 MHz, CDCl_3_) δ 8.10–7.99 (m, 2H, 2 × CH_ar_), 7.59–7.43 (m, 3H, 3 × CH_ar_), 7.42–7.28 (m, 3H), 7.38 (d, ^3^*J*_HH_ = 15.8 Hz, 1H), 6.78 (s, 1H, NH), 5.73 (d, ^3^*J*_HH_ = 15.8 Hz, 1H, CH=), 4.52 (s, 2H, CH_2_), 1.42 (s, 9H, 3 × CH_3_*t*Bu) ppm. ^13^C NMR {^1^H} (75 MHz, CDCl_3_) δ 166.1 (C=O), 165.4 (C=O), 140.4 (C_quat_), 139.3 (C_quat_), 134.1 (CH=), 132.9 (C_quat_), 132.2 (q, ^2^*J*_FC_ = 32.3 Hz, C_quat_), 132.0 (q, ^2^*J*_FC_ = 32.7 Hz, C_quat_), 130.2 (2 × CH_Ar_), 124.0 (q, ^1^*J*_FC_ = 272.4 Hz), 123.9 (q, ^1^*J*_FC_ = 272.7 Hz), 124.1 (CH_Ar_), 121.9 (CH_Ar_), 121.6 (q, ^3^*J*_FC_ = 3.9 Hz, CH_Ar_), 120.9 (q, ^3^*J*_FC_ = 3.9 Hz, CH_Ar_), 119.4 (CH=), 117.5 (q, ^3^*J*_FC_ = 3.9 Hz, CH_Ar_), 115.3 (q, ^3^*J*_FC_ = 3.9 Hz, CH_Ar_), 112.1 (C_quat_), 81.0 (C_quat_*t*Bu), 49.2 (CH_2_), 28.22 (3 × CH_3_*t*Bu) ppm. ^19^F-NMR (282 MHz, CDCl_3_) δ −63.1, −63.3 ppm. FTIR (neat) ν_max_: 3301 (NH _st_), 3051 (=CH _st_), 1708 (C=O _st_), 1675 (C=O _st_), 1607 (C=C _st_) cm^−1^. HRMS (ESI-TOF) *m/z*: [M+H]^+^ calcd for C_25_H_23_F_6_N_2_O_3_ 513.1613, Found 513.1610. 

Naphthalen-2-yl (*E*)-3-(5-oxo-1-(*p*-tolyl)-4-(*p*-tolylamino)-2,5-dihydro-1*H*-pyrrol-3-yl) acrylate (**7f**). The general procedure A was followed using 1-(*p*-tolyl)-3-(*p*-tolylamino)-1,5-dihydro-2*H*-pyrrol-2-one **4a** (278 mg, 1 mmol) and naphthalen-2-yl prop-2-ynoate (235 mg, 1.2 mmol), affording 450 mg (95%) of **7f** as brown crystals after crystallization (Dichloromethane/Hexanes 1:3). M.p. (Dichloromethane/Hexanes) = 228–230 °C. (dec.). ^1^H NMR (400 MHz, CDCl_3_) δ 7.84 (d, ^3^*J*_HH_ = 8.4 Hz, 2H, 2 × CH_Ar_), 7.75 (d, ^3^*J*_HH_ = 15.6 Hz, 1H, CH=), 7.70 (d, ^3^*J*_HH_ = 8.6 Hz, 2H, 2 × CH_Ar_), 7.58–7.35 (m, 5H, 5 × CH_Ar_), 7.25–7.02 (m, 6H, 6 × CH_Ar_), 6.86 (s, 1H, NH), 5.84 (d, ^3^*J*_HH_ = 15.6 Hz, 1H, CH=), 4.50 (s, 2H, CH_2_), 2.38 (s, 3H, CH_3_Tol), 2.30 (s, 3H, CH_3_Tol) ppm. ^13^C NMR {^1^H} (101 MHz, CDCl_3_) δ 165.5 (C=O), 165.4 (C=O), 145.7 (C_quat_), 137.4 (C_quat_), 137.2 (CH=), 136.2 (C_quat_), 135.8 (C_quat_), 135.0 (C_quat_), 134.8 (C_quat_), 133.9 (C_quat_), 131.5 (C_quat_), 130.3 (2 × CH_Ar_), 130.0 (2 × CH_Ar_), 127.9 (CH_Ar_), 127.7 (CH_Ar_), 126.6 (CH_Ar_), 126.5 (CH_Ar_), 125.7 (CH_Ar_), 123.2 (2 × CH_Ar_), 121.3 (CH_Ar_), 119.2 (2 × CH_Ar_), 118.5 (CH_Ar_), 112.0 (CH=), 106.8 (C_quat_), 49.3 (CH_2_), 21.2 (CH_3_Tol), 21.1 (CH_3_Tol) ppm. FTIR (neat) ν_max_: 3284 (NH _st_), 3056 (=CH _st_), 1714 (C=O _st_), 1676 (C=O _st_), 1603 (=CH _st_) cm^−1^. HRMS (ESI-TOF) *m/z*: [M+H]^+^ calcd for C_31_H_26_N_2_O_3_ 475.2022, Found 475.2024. 

(*E*)-4-(3-oxobut-1-en-1-yl)-1-(*p*-tolyl)-3-(*p*-tolylamino)-1,5-dihydro-2*H*-pyrrol-2-one (**7g**). The general procedure B was followed using 1-(*p*-tolyl)-3-(*p*-tolylamino)-1,5-dihydro-2*H*-pyrrol-2-one (278 mg, 1 mmol) **4a**, affording 312 mg (90%) of **7g** as yellow crystals after crystallization (Et_2_O/pentane 1:3). M.p. (Et_2_O /Pentane) = 195–196 °C. ^1^H NMR (300 MHz, CDCl_3_) δ 7.66 (d, ^3^*J*_HH_ = 8.5 Hz, 2H, 2 × CH_Ar_), 7.36–7.10 (m, 7H, 6 × CH_Ar_ + CH=), 6.81 (s, 1H, NH), 5.86 (d, ^3^*J*_HH_ = 16.1 Hz, 1H, CH=), 4.41 (s, 2H, CH_2_), 2.38 (s, 3H, CH_3_Tol), 2.36 (s, 3H, CH_3_Tol), 2.01 (d, ^4^*J*_HH_ = 1.5 Hz, 3H, COCH_3_) ppm. ^13^C NMR {^1^H} (75 MHz, CDCl_3_) δ 198.1 (C=O), 165.3 (C=O), 137.6 (C_quat_), 136.6 (C_quat_), 136.2 (C_quat_), 135.8 (C_quat_), 134.9 (C_quat_), 134.7 (CH=), 130.2 (2 × CH_Ar_), 129.9 (2 × CH_Ar_), 123.5 (2 × CH_Ar_), 119.1 (2 × CH_Ar_), 119.1 (CH=), 107.3 (C_quat_), 49.1 (CH_2_), 26.0 (COCH_3_), 21.1 (CH_3_Tol), 21.1 (CH_3_Tol) ppm. FTIR (neat) ν_max_: 3281 (NH _st_), 3059 (=CH _st_), 1713 (C=O _st_), 1679 (C=O _st_), 1603 (C=C _st_) cm^−1^. HRMS (ESI-TOF) *m/z*: [M+H]^+^ calcd for C_22_H_23_N_2_O_2_ 347.1760, Found 347.1757. 

(*E*)-4-(3-oxobut-1-en-1-yl)-1-(3-(trifluoromethyl)phenyl)-3-((3-(trifluoromethyl)phenyl) amino)-1,5-dihydro-2*H*-pyrrol-2-one (**7h**). The general procedure B was followed using 1-(3-(trifluoromethyl)phenyl)-3-((3-(trifluoromethyl)phenyl)amino)-1,5-dihydro-2*H*-pyrrol-2-one **4b** (386 mg, 1 mmol), affording 391 mg (86%) of **7h** as a pale yellow oil after chromatography (Hexanes/AcOEt 8:2). ^1^H NMR (400 MHz, Acetone *d*_6_) δ 8.42 (m, 1H, CH_Ar_), 8.22 (m, 1H, CH_Ar_), 8.05 (s, 1H, NH), 7.67 (m, 1H, CH_Ar_), 7.62–7.49 (m, 4H, 4 × CH_Ar_), 7.45 (d, ^3^*J*_HH_ = 16.1 Hz, 1H, CH=), 7.39 (m, 1H, CH_Ar_), 6.3 (d, ^3^*J*_HH_ = 16.1 Hz, 1H, CH=), 4.83 (s, 2H, CH_2_), 2.10 (s, 3H, COCH_3_) ppm. ^13^C NMR {^1^H} (101 MHz, Acetone *d*_6_) δ 197.1 (C=O), 165.5 (C=O), 143.0 (C_quat_), 141.0 (C_quat_), 136.6 (C_quat_), 133.1 (CH=), 131.7 (q, ^2^*J*_FC_ = 32.1 Hz, C_quat_), 131.6 (q, ^2^*J*_FC_ = 32.2 Hz, C_quat_), 130.9 (2 × CH_Ar_), 127.1 (CH_Ar_), 125.2 (q, ^1^*J*_FC_ = 271.4 Hz, C_quat_), 125.2 (q, ^1^*J*_FC_ = 271.6 Hz, C_quat_), 124.5 (CH_Ar_), 122.4 (CH=), 121.2 (q, ^3^*J*_FC_ = 3.8 Hz, CH_Ar_), 119.9 (q, ^3^*J*_FC_ = 3.7 Hz, CH_Ar_), 117.7 (C_quat_), 117.4 (q, ^3^*J*_FC_ = 4.2 Hz, CH_ar_), 115.6 (q, ^3^*J*_FC_ = 3.8 Hz, CH_Ar_), 49.6 (CH_2_), 26.8 (COCH_3_) ppm. ^19^F-NMR (282 MHz, CDCl_3_) δ −63.1, −63.5 ppm. FTIR (neat) ν_max_: 3297 (NH _st_), 3054 (=CH _st_), 1715 (C=O _st_), 1677 (C=O _st_), 1608 (C=C _st_) cm^−1^. HRMS (ESI-TOF) *m/z*: [M+H]^+^ calcd for C_22_H_17_F_6_N_2_O_2_ 455.1194, Found 455.1187.

##### Procedure for the Catalytic Hydrogenation of **7a**

A solution of (*E*)-3-(5-oxo-1-(*p*-tolyl)-4-(*p*-tolylamino)-2,5-dihydro-1*H*-pyrrol-3-yl) acrylate (**7a**) (363 mg, 1mmol) in MeOH (5 mL) was stirred for 2 days under H_2_ pressure (80 psi) in the presence of palladium on carbon (10%) (213 mg, 10% mmol). The reaction mixture was filtered through celite and concentrated under vacuum. The crude residue was purified by crystallization (Dichloromethane/Hexanes 1:3), affording pure saturated γ-lactam **7**.

Methyl 3-((3*R**,4*R**)-5-oxo-1-(*p*-tolyl)-4-(*p*-tolylamino)pyrrolidin-3-yl)propanoate (**8**). The procedure was followed, affording 327 mg (90%) of **8** as white crystals M.p. (Dichloromethane/Hexanes) = 158–160 °C. ^1^H NMR (400 MHz, CDCl_3_) δ 7.54 (d, ^3^*J*_HH_ = 8.5 Hz, 2H, 2 × CH_Ar_), 7.21 (d, ^3^*J*_HH_ = 8.5 Hz, 2H, 2 × CH_Ar_), 7.04 (d, ^3^*J*_HH_ = 8.4 Hz, 2H, 2 × CH_Ar_), 6.63 (d, ^3^*J*_HH_ = 8.4 Hz, 2H, 2 × CH_Ar_), 4.47 (s, 1H, NH), 4.13 (dd, ^3^*J*_HH_ = 7.2 Hz, ^4^*J*_HH_ = 2.0 Hz, 1H, NCH), 4.00 (dd, ^2^*J*_HH_ = 10.2 Hz, ^3^*J*_HH_ = 5.8 Hz, 1H, NCH_A_H_B_), 3.59 (s, 3H, OCH_3_), 2.91 (m, 1H, CH), 2.43–2.29 (m, 5H, CH_3_Tol + COCH_2_), 2.27 (s, 3H, CH_3_Tol), 1.96 (m, 1H, CH), 1.54 (m, 1H, CH_A_H_B_) ppm. ^13^C NMR {^1^H} (101 MHz, CDCl_3_) δ 173.5 (C=O), 171.4 (C=O), 145.4 (C_quat_), 136.7 (C_quat_), 134.9 (C_quat_), 130.0 (2 × CH_Ar_), 129.7 (2 × CH_Ar_), 127.8 (C_quat_), 119.8 (2 × CH_Ar_), 113.5 (2 × CH_Ar_), 60.0 (CH), 51.7 (OCH_3_), 50.1 (CH_2_), 37.1 (CH), 31.7 (CH_2_), 23.0 (CH_2_), 21.0 (CH_3_Tol), 20.5 (CH_3_Tol) ppm. FTIR (neat) ν_max_: 3325 (NH _st_), 1733 (C=O _st_), 1698 (C=O _st_) cm^−1^. HRMS (ESI-TOF) *m/z*: [M+H]^+^ calcd for C_22_H_27_N_2_O_3_ 367.2022, Found 367.2011.

##### Procedure for the Intramolecular Cyclization of **8**

To a suspension of NaH (29 mg, 1.2 mmol) in toluene (3 mL) was added neat 3-((3*R**,4*R**)-5-oxo-1-(*p*-tolyl)-4-(*p*-tolylamino)pyrrolidin-3-yl)propanoate (**8**) (366 mg, 1mmol). The reaction was stirred under reflux overnight and the resulting mixture was then quenched with water (10 mL) and extracted with dichloromethane (2 × 10 mL). The combined organic layers were dried over MgSO_4_ and concentrated under vacuum and the crude residue was purified by chromatography (Hexanes/AcOEt 1:1), affording pure bicyclic γ-lactam **9**.

(4a*R**,7a*R**)-1,6-Di-*p*-tolylhexahydro-1*H*-pyrrolo[3,4-b]pyridine-2,7-dione (**9**). The procedure was followed, affording 330 mg (98%) of **9** as a pale yellow oil. ^1^H NMR (400 MHz, CDCl_3_) δ 7.51 (d, ^3^*J*_HH_ = 8.6 Hz, 2H, 2 × CH_Ar_), 7.37 (d, ^3^*J*_HH_ = 8.4 Hz, 2H, 2 × CH_Ar_), 7.19 (d, ^3^*J*_HH_ = 8.4 Hz, 2H, 2 × CH_Ar_), 7.17 (d, ^3^*J*_HH_ = 8.6 Hz, 2H, 2 × CH_Ar_), 4.65 (d, ^3^*J*_HH_ = 8.0 Hz, 1H, NCH), 4.04 (dd, ^2^*J*_HH_ = 10.1 Hz, ^3^*J*_HH_ = 6.4 Hz, 1H, NCH_A_CH_B_), 3.56 (d, ^2^*J*_HH_ = 10.1 Hz, 1H, NCH_A_CH_B_), 3.00 (p, ^3^*J*_HH_ = 7.0 Hz, 1H, CH), 2.54 (t, ^3^*J*_HH_ = 6.7 Hz, 2H, COCH_2_), 2.34 (s, 3H, CH_3_Tol), 2.33 (s, 3H, CH_3_Tol), 2.16–1.92 (m, 2H, COCH_2_CH_2_) ppm. ^13^C NMR {^1^H} (101 MHz, CDCl_3_) δ 170.4 (C=O), 169.7 (C=O), 139.7 (C_quat_), 136.9 (C_quat_), 136.4 (C_quat_), 134.9 (C_quat_), 129.6 (2 × CH_Ar_), 129.5 (2 × CH_Ar_), 127.3 (2 × CH_Ar_), 119.5 (2 × CH_Ar_), 64.3 (CH), 51.7 (CH_2_), 31.5 (CH_2_), 30.3 (CH), 25.7 (CH_2_), 21.2 (CH_3_Tol), 20.9 (CH_3_Tol) ppm. FTIR (neat) ν_max_: 1698 (C=O _st_), 1673 (C=O _st_) cm^−1^. HRMS (ESI-TOF) *m/z*: [M+H]^+^ calcd for C_21_H_23_N_2_O_2_ 335.1760, Found 335.1757. 

##### Procedure for the Acidic Hydrolysis of **7a**

A solution of (*E*)-3-(5-oxo-1-(*p*-tolyl)-4-(*p*-tolylamino)-2,5-dihydro-1*H*-pyrrol-3-yl) acrylate (**7a**) (0.363, 1 mmol) in toluene (2 mL) and 37% aqueous HCl (2 mL) was stirred at room temperature overnight. The mixture was diluted with water and extracted with dichloromethane (2 × 10 mL). The combined organic layers were washed with water (2 × 10 mL), dried over MgSO_4_, and concentrated under vacuum. The crude residue was purified by chromatography (Hexanes/AcOEt 4:1), affording pure enol-derived γ-lactam **10** after crystallization (Et_2_O /Pentane 1:3).

Methyl (*E*)-3-(4-hydroxy-5-oxo-1-(*p*-tolyl)-2,5-dihydro-1*H*-pyrrol-3-yl)acrylate (**10**). The procedure was followed, affording 262 mg (96%) of **9** as white crystals. M.p. (Et_2_O /Pentane) = 239–241 °C. ^1^H NMR (400 MHz, DMSO-*d*6) δ 11.33 (s, 1H, OH), 7.72 (d, ^3^*J*_HH_ = 8.7 Hz, 2H, 2 × CH_Ar_), 7.66 (d, ^3^*J*_HH_ = 15.9 Hz, 1H, CH=), 7.22 (d, ^3^*J*_HH_ = 8.7 Hz, 2H, 2 × CH_Ar_), 6.11 (d, ^3^*J*_HH_ = 15.9 Hz, 1H, CH=), 4.49 (s, 2H, CH_2_), 3.71 (s, 3H, OCH_3_), 2.29 (s, 3H, CH_3_Tol) ppm. ^13^C NMR {^1^H} (101 MHz, DMSO-*d*6) δ 167.1 (C=O), 164.6 (C=O), 149.6 (C_quat_), 137.2 (C_quat_), 133.9 (CH=), 133.8 (C_quat_), 129.8 (2 × CH_Ar_), 118.8 (2 × CH_Ar_), 116.9 (CH=), 114.9 (C_quat_), 52.0 (OCH_3_), 47.2 (CH_2_), 20.9 (CH_3_Tol) ppm. FTIR (neat) ν_max_: 3460 (OH _st_), 3046 (=CH _st_), 1701 (C=O _st_), 1679 (C=O _st_), 1612 (C=C _st_) cm^−1^. HRMS (ESI-TOF) *m/z*: [M+H]^+^ calcd for C_15_H_16_NO_4_ 274.1079, Found 274.1076.

##### General Procedure for the Nucleophilic Addition of γ-Lactams **4c**–**d** to Ethyl Glyoxalate

A solution of diethyl (5-oxo-1-(*p*-tolyl)-4-(*p*-tolylamino)-2,5-dihydro-1*H*-pyrrol-3-yl) phosphonate **4c** (414 mg, 1 mmol) in THF (1 mL) was added to a freshly prepared solution of LDA (1.2 Eq) in THF (2 mL) at −78 °C. After 1 h, a 1.6 M solution of ethyl glyoxalate in toluene (1.2 mmol, 750 μL) was added and the reaction was left to warm to rt overnight. The reaction was quenched with a 0.5 M aqueous solution of HCl and the resulting mixture was extracted with dichloromethane (2 × 10 mL). The combined organic layers were washed with water (2 × 10 mL), dried over MgSO_4_, and concentrated under vacuum. The crude residue was purified by chromatography (Hexanes/AcOEt 6:4), affording pure γ-lactam derivatives **12**.

Ethyl (*R**)-2-((*R**)-3-(diethoxyphosphoryl)-5-oxo-1-(*p*-tolyl)-4-(*p*-tolylamino)-2,5-dihydro-1*H*-pyrrol-2-yl)-2-hydroxyacetate (**12a**). The general procedure was followed using diethyl (5-oxo-1-(*p*-tolyl)-4-(*p*-tolylamino)-2,5-dihydro-1*H*-pyrrol-3-yl)phosphonate **4c** (414 mg, 1 mmol) affording 315 mg (61%) of **12a** as a pale yellow oil. ^1^H NMR (400 MHz, CDCl_3_) δ 7.45 (s, 1H, NH), 7.34 (d, ^3^*J*_HH_ = 7.3 Hz, 2H, 2 × CH_Ar_), 7.20 (d, ^3^*J*_HH_ = 8.0 Hz, 2H, 2 × CH_Ar_), 7.14–7.03 (m, 4H, 4 × CH_Ar_), 5.16 (t, ^3^*J*_HH_ = 1.5 Hz, 1H, CH), 4.47 (d, ^3^*J*_PH_ = 7.9 Hz, 1H, CH), 4.42 (broad s, 1H, OH), 4.03 (q, ^3^*J*_HH_ = 7.2 Hz, 4H, 2 × CH_2_), 3.94 (q, ^3^*J*_HH_ = 7.2 Hz, 2H, CH_2_), 2.35 (s, 3H, CH_3_Tol), 2.32 (s, 3H, CH_3_Tol), 1.47–0.98 (m, 9H, 3 × CH_3_) ppm. ^13^C NMR {^1^H} (101 MHz, CDCl_3_) δ 171.5 (C=O), 164.1 (d, ^2^*J*_CP_ = 19.4 Hz, C_quat_), 146.2 (d, ^3^*J*_CP_ = 7.2 Hz, C=O), 137.0 (C_quat_), 136.5 (C_quat_), 135.0 (C_quat_), 133.1 (C_quat_), 129.8 (2 × CH_Ar_), 129.3 (2 × CH_Ar_), 124.4 (2 × CH_Ar_), 123.8 (2 × CH_Ar_), 96.4 (d, ^1^*J*_CP_ = 211.3 Hz, C_quat_), 70.2 (CH), 63.9 (d, ^2^*J*_CP_ = 17.4 Hz, CH), 62.5 (d, ^2^*J*_CP_ = 5.5 Hz, CH_2_), 62.4 (d, ^2^*J*_CP_ = 5.4 Hz, CH_2_), 61.8 (CH_2_), 21.2 (CH_3_Tol), 21.1 (CH_3_Tol), 16.3 (d, ^3^*J*_CP_ = 6.7 Hz, CH_3_), 16.3 (d, ^3^*J*_CP_ = 6.9 Hz, CH_3_), 14.1 (CH_3_) ppm. ^31^P NMR (162 MHz, CDCl_3_) δ 19.1 ppm. FTIR (neat) ν_max_: 3430 (OH), 3338 (NH), 1714 (C=O _st_), 1694 (C=O _st_), 1609 (C=C), 1213 (P=O) cm^−1^. HRMS (ESI-TOF) *m/z*: [M+H]^+^ calcd for C_26_H_34_N_2_O_7_P 517.2104, Found 517.2100. 

Ethyl (*R**)-2-((*R**)-3-(diphenylphosphoryl)-5-oxo-1-(*p*-tolyl)-4-(*p*-tolylamino)-2,5-dihydro-1*H*-pyrrol-2-yl)-2-hydroxyacetate (**12b**). The general procedure was followed using 4-(diphenylphosphoryl)-1-(*p*-tolyl)-3-(*p*-tolylamino)-1,5-dihydro-2*H*-pyrrol-2-one **4d** (478 mg, 1 mmol), affording 498 mg (86%) of **12b** as yellow crystals. M.p. (Dichloromethane/Hexanes) = 211–213 °C. ^1^H NMR (400 MHz, CDCl_3_) δ 7.87–7.70 (m, 2H, 2 × CH_Ar_), 7.61–7.43 (m, 5H, 5 × CH_Ar_), 7.33 (d, ^3^*J*_HH_ = 8.2 Hz, 2H, 2 × CH_Ar_), 7.31–7.23 (m, 3H, 3 × CH_Ar_), 7.17 (d, ^3^*J*_HH_ = 8.2 Hz, 2H, 2 × CH_Ar_), 6.82 (d, ^3^*J*_HH_ = 8.2 Hz, 2H, 2 × CH_Ar_), 6.72 (d, ^3^*J*_HH_ = 8.2 Hz, 2H, 2 × CH_Ar_), 6.64 (s, 1H, NH), 5.28 (broad s, 1H, OH), 5.09 (t, ^3^*J*_HH_ = 2.8 Hz, 1H, CH), 4.38 (dd, ^3^*J*_PH_ = 9.3 Hz, ^3^*J*_HH_ = 2.8 Hz, 1H, CH), 3.91 (q, ^3^*J*_HH_ = 7.1 Hz, 2H, CH_2_), 2.33 (s, 3H, CH_3_Tol), 2.21 (s, 3H, CH_3_Tol), 1.12 (t, ^3^*J*_HH_ = 7.1 Hz, 3H, CH_3_) ppm. ^13^C NMR {^1^H} (101 MHz, CDCl_3_) δ 171.0 (C=O), 164.5 (d, ^3^*J*_CP_ = 14.5 Hz, C=O), 144.9 (d, ^2^*J*_CP_ = 3.9 Hz, C_quat_), 137.2 (C_quat_), 136.7 (d, ^4^*J*_CP_ = 1.2 Hz, C_quat_), 133.8 (C_quat_), 133.3 (C_quat_), 132.9 (d, ^1^*J*_CP_ = 109.0 Hz, C_quat_), 132.1 (d, ^4^*J*_CP_ = 2.5 Hz, CH_Ar_), 131.9 (d, ^4^*J*_CP_ = 2.9 Hz, CH_Ar_), 131.3 (d, ^2^*J*_CP_ = 9.6 Hz, 2 × CH_Ar_), 131.2 (d, ^2^*J*_CP_ = 10.3 Hz, 2 × CH_Ar_), 131.0 (C_quat_), 129.7 (2 × CH_Ar_), 129.5 (2 × CH_Ar_), 129.0 (d, ^3^*J*_CP_ = 12.4 Hz, 2 × CH_Ar_), 128.6 (d, ^3^*J*_CP_ = 12.7 Hz, 2 × CH_Ar_), 123.7 (2 × CH_Ar_), 121.5 (2 × CH_Ar_), 102.5 (d, ^1^*J*_CP_ = 113.1 Hz, C_quat_), 70.6 (CH), 64.7 (d, ^2^*J*_CP_ = 12.7 Hz, CH), 61.6 (CH_2_), 21.2 (CH_3_Tol), 20.9 (CH_3_Tol), 14.0 (CH_3_) ppm. ^31^P NMR (121 MHz, CDCl_3_) δ 19.4 ppm. FTIR (neat) ν_max_: 3417 (OH), 3262 (NH), 1720 (C=O), 1679 (C=O), 1606 (C=C), 1175 (P=O). HRMS (ESI-TOF) *m/z*: [M+H]^+^ calcd for C_34_H_34_N_2_O_5_P 581.2205, Found 581.2202. 

##### Procedure for the Synthesis of γ-Lactam Derivative **13**

A solution of 4-(diphenylphosphoryl)-1-(*p*-tolyl)-3-(*p*-tolylamino)-1,5-dihydro-2*H*-pyrrol-2-one (**4d**) (478 mg, 1 mmol) in THF (1 mL) was added to a freshly prepared solution of LDA (1.2 Eq) in THF (2 mL) at −78 °C. After 1 h, a solution of 1-isocyanato-4-nitrobenzene (197 mg, 1.2 mmol) in THF (1 mL) was added and the reaction was left to warm to rt overnight. The reaction was quenched with a 0.5 M aqueous solution of HCl and the resulting mixture was extracted with dichloromethane (2 × 10 mL). The combined organic layers were washed with water (2 × 10 mL), dried over MgSO_4_, and concentrated under vacuum. The crude residue was purified by chromatography (Hexanes/AcOEt 4:1), affording pure γ-lactam derivative **13**.

4-(Diphenylphosphoryl)-5-((4-nitrophenyl)imino)-1-(*p*-tolyl)-3-(*p*-tolylamino)-1,5-dihydro-2*H*-pyrrol-2-one (**13**). The procedure was followed, affording 380 mg (62%) of **13** as an orange oil. ^1^H NMR (400 MHz, CDCl_3_) δ 10.54 (s, 1H, NH), 8.03–7.91 (m, 4H, 4 × CH_Ar_), 7.66 (d, ^3^*J*_HH_ = 8.9 Hz, 2H, 2 × CH_Ar_), 7.63–7.56 (m, 2H, 2 × CH_Ar_), 7.54–7.47 (m, 4H, 4 × CH_Ar_), 7.18 (d, ^3^*J*_HH_ = 8.3 Hz, 2H, 2 × CH_Ar_), 7.11 (d, ^3^*J*_HH_ = 8.2 Hz, 2H, 2 × CH_Ar_), 6.76 (d, ^3^*J*_HH_ = 8.2 Hz, 2H, 2 × CH_Ar_), 6.66 (d, ^3^*J*_HH_ = 8.3 Hz, 2H, 2 × CH_Ar_), 6.27 (d, ^3^*J*_HH_ = 8.9 Hz, 2H, 2 × CH_Ar_), 2.32 (s, 3H, CH_3_Tol), 2.10 (s, 3H, CH_3_Tol) ppm. ^13^C NMR {^1^H} (101 MHz, CDCl_3_) δ 164.7 (d, ^2^*J*_CP_ = 14.4 Hz, C=N), 153.2 (C_quat_), 152.1 (d, ^3^*J*_CP_ = 3.7 Hz, C=O), 148.7 (C_quat_), 148.6 (C_quat_), 142.5 (C_quat_), 138.7 (C_quat_), 136.4 (C_quat_), 134.7 (C_quat_), 133.4 (C_quat_), 132.5 (d, ^4^*J*_PC_ = 2.7 Hz, 2 × CH_Ar_), 132.2 (2 × CH_Ar_), 132.1 (2 × CH_Ar_), 130.7 (C_quat_), 129.4 (2 × CH_Ar_), 129.3 (2 × CH_Ar_), 128.5 (2 × CH_Ar_), 128.4 (2 × CH_Ar_), 128.1 (2 × CH_Ar_), 124.6 (2 × CH_Ar_), 123.8 (2 × CH_Ar_), 120.6 (2 × CH_Ar_), 93.7 (d, ^1^*J*_CP_ = 116.4 Hz, C_quat_), 21.2 (CH_3_Tol), 21.1 (CH_3_Tol) ppm. ^31^P NMR (162 MHz, CDCl_3_) δ 30.9 ppm. FTIR (neat) ν_max_: 3330 (NH), 1667 (C=O _st_), 1606 (C=C), 1204 (P=O) cm^−1^. HRMS (ESI-TOF) *m/z*: [M+H]^+^ calcd for C_36_H_30_N_4_O_4_P 613.2005, Found 613.2025.

##### Procedure for the Functionalization of γ-Lactam **5a** with Methyl Iodide

A solution of 5-phenyl-1-(*p*-tolyl)-3-(*p*-tolylamino)-1,5-dihydro-2*H*-pyrrol-2-one (**5a**) (354 mg, 1mmol) in THF (1 mL) was added to a solution of freshly prepared LDA (1.2 mmol) in tetrahydrofuran (2 mL) at −78 °C. After 1 h, an excess of methyl iodide (93 μL, 1.5 mmol) was added. The reaction mixture was stirred overnight at −78 °C and was then quenched with a 0.5 M aqueous solution of HCl (10 mL). The reaction mixture was extracted with dichloromethane (2 × 10 mL) and the combined organic layers were washed with water (2 × 10 mL), dried with MgSO_4_, and concentrated under vacuum. The crude residue was purified by chromatography (Hexanes), affording a 3:1 mixture of **18** and **19** that was isolated as pure after chromatography followed by crystallization (Et_2_O/Pentane).

4-methyl-5-phenyl-1-(*p*-tolyl)-3-(*p*-tolylamino)-1,5-dihydro-2*H*-pyrrol-2-one (**18**). The procedure was followed, affording 0.188 g (51%) of **18** as white crystals. M.p. (Et_2_O/Pentane) = 191–192 °C. ^1^H NMR (400 MHz, CDCl_3_) δ 7.40 (d, ^3^*J*_HH_ = 8.5 Hz, 2H, 2 × CH_Ar_), 7.35–7.20 (m, 5H, 5 × CH_Ar_), 7.05 (d, ^3^*J*_HH_ = 8.5 Hz, 2H, 2 × CH_Ar_), 7.05 (d, ^3^*J*_HH_ = 8.5 Hz, 2H, 2 × CH_Ar_), 6.78 (d, ^3^*J*_HH_ = 8.5 Hz, 2H, 2 × CH_Ar_), 5.94 (s, 1H, NH), 5.33 (s, 1H, CH), 2.28 (s, 3H, CH_3_Tol), 2.24 (s, 3H, CH_3_Tol), 1.60 (s, 3H, CH_3_) ppm. ^13^C NMR {^1^H} (101 MHz, CDCl_3_) δ 168.2 (C=O), 139.7 (C_quat_), 137.3 (C_quat_), 135.2 (C_quat_), 134.2 (C_quat_), 130.8 (C_quat_), 129.7 (C_quat_), 129.6 (2 × CH_Ar_), 129.5 (2 × CH_Ar_), 129.1 (2 × CH_Ar_), 128.4 (CH_Ar_), 127.2 (2 × CH_Ar_), 125.7 (C_quat_), 121.4 (2 × CH_Ar_), 118.8 (2 × CH_Ar_), 67.6 (CH), 21.0 (CH_3_Tol), 20.8 (CH_3_Tol), 13.6 (CH_3_) ppm. FTIR (neat) ν_max_: 3306 (NH _st_), 1688 (C=O _st_), 1608 (C=C _st_) cm^−1^. HRMS (ESI-TOF) *m/z*: [M+H]^+^ calcd for C_25_H_25_N_2_O 369.1967, Found 369.1973.

3-Hydroxy-4-methyl-5-phenyl-1-(*p*-tolyl)-1,5-dihydro-2*H*-pyrrol-2-one (**19**). The procedure was followed, affording 76 mg (26%) of **19** of as a pale yellow oil. ^1^H NMR (400 MHz, CDCl_3_) δ 7.34 (d, ^3^*J*_HH_ = 8.5 Hz, 2H, 2 × CH_Ar_), 7.33–7.24 (m, 3H, 3 × CH_Ar_), 7.16 (d, ^3^*J*_HH_ = 8.3 Hz, 2H, 2 × CH_Ar_), 7.04 (d, ^3^*J*_HH_ = 8.5 Hz, 2H, 2 × CH_Ar_), 6.15 (s, 1H, OH), 5.26 (s, 1H, CH), 2.23 (s, 3H, CH_3_Tol), 1.73 (s, 3H, CH_3_) ppm. ^13^C NMR {^1^H} (101 MHz, CDCl_3_) δ 166.9 (C=O), 140.7 (C_quat_), 136.2 (C_quat_), 134.8 (C_quat_), 134.6 (C_quat_), 129.6 (2 × CH_Ar_), 129.1 (2 × CH_Ar_), 128.5 (CH_Ar_), 127.3 (2 × CH_Ar_), 122.1 (C_quat_), 121.3 (2 × CH_Ar_), 66.0 (CH), 21.0 (CH_3_Tol), 9.5 (CH_3_) ppm. FTIR (neat) ν_max_: 3501 (OH _st_), 1681 (C=O _st_), 1607 (C=C _st_) cm^−1^. HRMS (ESI-TOF) *m/z*: [M+H]^+^ calcd for C_18_H_18_N_2_O 280.1338, Found 280.1343. Additionally, substrate **18** can be obtained in 95% yield from **17** by refluxing in a mixture of THF 1 (mL) and a 0.1 M aqueous solution of HCl (1 mL) for 1 h.

##### Procedure for the Vinylogous Mannich Reaction of γ-Lactam **5b** with Ethyl Glyoxalate

A solution of ethyl 5-oxo-1-(*p*-tolyl)-4-(*p*-tolylamino)-2,5-dihydro-1*H*-pyrrole-2-carboxylate (**5b**)(350 mg, 1 mmol), *N*,*N*-dimethylmethyleneiminium iodide (370 mg, 2 mmol) and triethylamine (350 μL, 2.5 mmol, in chloroform (3 mL) was refluxed overnight. The reaction mixture was then acidified with a 0.5 M aqueous solution of HCl (10 mL) and extracted with dichloromethane (2 × 10 mL). The combined organic layers were washed with water (2 × 10 mL), dried over MgSO_4_, and concentrated under vacuum. The crude residue was purified by crystallization from a mixture of dichloromethane/hexanes (1:3), affording the pure functionalized γ-lactam derivative **20**.

Ethyl 2-((dimethylamino)methyl)-5-oxo-1-(*p*-tolyl)-4-(*p*-tolylamino)-2,5-dihydro-1*H*- pyrrole-2-carboxylate (**20**). The procedure was followed, affording 321 mg (68%) of **20** as yellow crystals. M.p. (Dichloromethane/Hexanes) = 170–172 °C. ^1^H NMR (400 MHz, CDCl_3_) δ 7.25 (d, ^3^*J*_HH_ = 8.5 Hz, 2H, 2 × CH_Ar_), 7.20 (d, ^3^*J*_HH_ = 8.2 Hz, 2H, 2 × CH_Ar_), 7.13 (d, ^3^*J*_HH_ = 8.2 Hz, 2H, 2 × CH_Ar_), 7.00 (d, ^3^*J*_HH_ = 8.5 Hz, 2H, 2 × CH_Ar_), 6.65 (s, 1H, NH), 5.95 (s, 1H, CH), 4.30–4.09 (m, 2H, CH_2_), 3.17 (d, ^2^*J*_HH_ = 14.0 Hz, 1H, CH_A_CH_B_), 2.74 (d, ^2^*J*_HH_ = 14.0 Hz, 1H, CH_A_CH_B_), 2.36 (s, 3H, CH_3_Tol), 2.31 (s, 3H, CH_3_Tol), 2.16 (s, 6H, 2 × NCH_3_), 1.24 (t, ^3^*J*_HH_ = 7.1 Hz, 3H, CH_3_) ppm. ^13^C NMR {^1^H} (101 MHz, CDCl_3_) δ 171.1 (C=O), 168.3 (C=O), 138.8 (C_quat_), 136.6 (C_quat_), 134.1 (C_quat_), 133.7 (C_quat_), 130.1 (C_quat_), 130.0 (2 × CH_Ar_), 129.7 (2 × CH_Ar_), 125.6 (2 × CH_Ar_), 116.9 (2 × CH_Ar_), 104.9 (CH_Ar_), 72.7 (CH), 62.1 (CH_2_O), 60.9 (CH_2_), 47.8 (2 × NCH_3_), 21.2 (CH_3_, Tol), 20.8 (CH_3_, Tol), 14.2 (CH_3_, OEt) ppm. FTIR (neat) ν_max_: 3312 (=CH _st_), 1691 (C=O _st_), 1615 (C=C _st_) cm^−1^. HRMS (ESI-TOF) *m/z*: [M+H]^+^-Me_2_N calcd for C_25_H_23_N_2_O 367.1801, Found 367.1806. 

##### Procedure for the Hydrogenation Reaction of γ-Lactam **5b**

A solution of ethyl 5-oxo-1-(*p*-tolyl)-4-(*p*-tolylamino)-2,5-dihydro-1*H*-pyrrole-2- carboxylate (**5b**) (350 mg, 1mmol) in EtOH (5 mL) was stirred for 2 days under H_2_ pressure (80 psi) in the presence of palladium on carbon (10%) (320 mg, 10% mmol). The reaction mixture was filtered through celite and concentrated under vacuum. The crude residue was purified by crystallization from a mixture of Et_2_O/Pentane 1:3, affording pure **21** as white crystals

Ethyl (2*S**,4*S**)-5-oxo-1-(*p*-tolyl)-4-(*p*-tolylamino)pyrrolidine-2-carboxylate (**21**). The procedure was followed, affording 326 mg (93%) of **21** as white crystals M.p. (Et_2_O/Pentane) = 142–144 °C. ^1^H NMR (400 MHz, CDCl_3_) δ 7.35 (d, ^3^*J*_HH_ = 8.5 Hz, 2H, 2 × CH_Ar_), 7.18 (d, ^3^*J*_HH_ = 8.2 Hz, 2H, 2 × CH_Ar_), 7.03 (d, ^3^*J*_HH_ = 8.2 Hz, 2H, 2 × CH_Ar_), 6.62 (d, ^3^*J*_HH_ = 8.5 Hz, 2H, 2 × CH_Ar_), 4.78 (dd, ^3^*J*_HH_ = 7.7 Hz, ^3^*J*_HH_ = 7.7 Hz, 1H, CH), 4.16 (dd, ^3^*J*_HH_ = 8.3 Hz, ^3^*J*_HH_ = 8.3 Hz, 1H, CH), 4.10 (m, 2H, CH_2_O), 3.09 (m, 1H, CH_2_), 2.33 (s, 3H, CH_3_Tol), 2.26 (s, 3H, CH_3_Tol), 2.08 (m,, 1H, CH_2_), 1.12 (m, 3H, CH_3_ OEt) ppm. ^13^C NMR {^1^H} (101 MHz, CDCl_3_) δ 172.4 (C=O), 171.0 (C=O), 144.7 (C_quat_), 136.1 (C_quat_), 135.2 (C_quat_), 130.0 (2 × CH_Ar_), 129.7 (2 × CH_Ar_), 128.1 (C_quat_), 122.1 (2 × CH_Ar_), 114.0 (2 × CH_Ar_), 61.9 (CH_2_), 59.1 (CH), 55.4 (CH), 33.1 (CH_2_), 21.1 (CH_3_Tol), 20.5 (CH_3_Tol), 14.1 (CH_3_) ppm. FTIR (neat) ν_max_: 3306 (N-H), 1748 (C=O), 1686 (C=O) cm^−1^. HRMS (ESI-TOF) *m/z*: [M+H]^+^ calcd for C_21_H_24_NO_3_ 353.1865, Found 353.1862.

##### Procedure for the [3+3] Annulation Reaction of γ-Lactam **5b** and β,γ-Unsaturated Ketoester **22**

A solution of 5-oxo-1-(*p*-tolyl)-4-(*p*-tolylamino)-2,5-dihydro-1*H*-pyrrole- 2-carboxylate (**5b**) (350 mg, 1 mmol), Yb(OTf)_3_ (62.0 mg, 0.1 mmol) and β,γ--unsaturated ketoester **22** (282 mg, 1.2 mmol) was stirred in CH_2_Cl_2_ at room temperature for 6 h. The resulting mixture was filtered through celite and concentrated under vacuum. The crude residue was purified by column chromatography (Hexanes/AcOEt (8:2), affording pure **23** as a white solid.

5-Ethyl 2-methyl (4*S**,5*S**)-4-(4-nitrophenyl)-7-oxo-1,6-di-*p*-tolyl-4,5,6,7-tetrahydro- 1*H*-pyrrolo[3,4-b]pyridine-2,5-dicarboxylate (**23**). The procedure was followed, affording 504 mg (89%) of **23** as a white solid. ^1^H NMR (400 MHz, CDCl_3_) δ 8.23 (d, ^3^*J*_HH_ = 8.7 Hz, 2H, 2 × CH_Ar_), 7.52 (d, ^3^*J*_HH_ = 8.7 Hz, 2H, 2 × CH_Ar_), 7.34 (d, ^3^*J*_HH_ = 8.2 Hz, 2H, 2 × CH_Ar_), 7.27 (d, ^3^*J*_HH_ = 8.0 Hz, 2H, 2 × CH_Ar_), 7.18 (d, ^3^*J*_HH_ = 8.0 Hz, 2H, 2 × CH_Ar_), 7.06 (d, ^3^*J*_HH_ = 8.2 Hz, 2H, 2 × CH_Ar_), 5.76 (d, ^3^*J*_HH_ = 4.7 Hz, 1H, CH), 5.03 (s, 1H, CH), 4.91 (d, ^3^*J*_HH_ = 4.7 Hz, 1H, CH), 3.49 (s, 3H, OCH_3_), 3.34 (m, 1H, CH_2_O), 3.11 (m, 1H, CH_2_O), 2.36 (s, 3H, CH_3_Tol), 2.25 (s, 3H, CH_3_Tol), 0.79 (dd, ^3^*J*_HH_ = 7.1 Hz, ^3^*J*_HH_ = 7.1 Hz, 3H, CH_3_ OEt) ppm. ^13^C NMR {^1^H} (101 MHz, CDCl_3_) δ 167.8 (C=O), 163.6 (C=O), 163.0 (C=O), 149.8 (C_quat_), 147.4 (C_quat_), 138.2 (C_quat_), 137.9 (C_quat_), 136.1 (C_quat_), 135.8 (C_quat_), 135.2 (C_quat_), 134.6 (C_quat_), 129.7 (2 × CH_Ar_), 129.2 (2 × CH_Ar_), 129.2 (2 × CH_Ar_), 128.6 (2 × CH_Ar_), 124.1 (2 × CH_Ar_), 121.0 (2 × CH_Ar_), 119.6 (C_quat_), 113.4 (CH), 63.1 (CH), 61.9 (CH_2_), 52.3 (OCH_3_), 41.0 (CH), 21.4 (CH_3_Tol), 20.8 (CH_3_Tol), 13.6 (CH_3_) ppm. FTIR (neat) ν_max_: 3053 (=CH _st_), 1739 (C=O), 1723 (C=O), 1676 (C=O), 1622 (C=C) cm^−1^. HRMS (ESI-TOF) *m/z*: [M+H]^+^ calcd for C_32_H_29_N_33_O_7_ 568.2084, Found 568.2078.

##### General Procedure for the Nucleophilic Addition of Organolithium Reagents to γ-Lactams **5**

A 1.6 M solution of the corresponding organolithium reagent in hexanes (1.25 mL, 2 mmol) was added to a solution of the corresponding γ-lactam **5** (1 mmol) in tetrahydrofuran (3 mL) at −78 °C under N_2_ atmosphere. The reaction was warmed to rt overnight. The reaction crude was quenched with a 0.5 M aqueous solution of HCl (5 mL), and extracted with dichloromethane (2 × 10 mL). The combined organic layers were washed with water, dried over MgSO_4_, and concentrated under reduced pressure. The crude residue was purified by crystallization, providing the pure functionalized γ-lactam derivatives **25**. In some cases, a previous purification by chromatography was necessary as detailed for each compound.

3-Methyl-5-phenyl-1-(*p*-tolyl)-3-(*p*-tolylamino)-1,3-dihydro-2*H*-pyrrol-2-one (**25a**). The general procedure was followed, using 5-phenyl-1-(*p*-tolyl)-3-(*p*-tolylamino)-1,5-dihydro- 2*H*-pyrrol-2-one (354 mg, 1 mmol) (**5a**) and a 1.6 M solution of methyllithium in hexanes (1.25 mL, 2 mmol), affording 288 mg (78%) of **25a** as yellow crystals after chromatography (Hexanes/AcOEt 8:2) and subsequent crystallization (Et_2_O /Pentane 1:3). M.p. (Et_2_O /Pentane) = 183–185 °C. ^1^H NMR (400 MHz, CDCl_3_) δ 7.31–7.25 (m, 4H, 4 × CH_Ar_), 7.24–7.20 (m, 2H, 2 × CH_Ar_), 7.11 (d, ^3^*J*_HH_ = 8.1 Hz, 2H, 2 × CH_Ar_), 7.03–6.95 (m, 6H, 5 × CH_Ar_ + NH), 5.63 (s, 1H, =CH), 2.33 (s, 3H, CH_3_Tol), 2.28 (s, 3H, CH_3_Tol), 1.45 (s, 3H, CH_3_) ppm. ^13^C NMR {^1^H} (101 MHz, CDCl_3_) δ 175.4 (C_quat_), 165.6 (C=O), 148.9 (C_quat_), 137.1 (C_quat_), 135.6 (C_quat_), 133.1 (C_quat_), 131.9 (C_quat_), 130.0 (CH_Ar_), 129.6 (2 × CH_Ar_), 129.4 (2 × CH_Ar_), 128.6 (2 × CH_Ar_), 128.3 (2 × CH_Ar_), 127.3 (2 × CH_Ar_), 121.8 (2 × CH_Ar_), 94.8 (=CH), 92.8 (C_quat_), 25.3 (CH_3_), 21.1 (CH_3_Tol), 21.1 (CH_3_Tol) ppm. FTIR (neat) ν_max_: 3403 (NH _st_), 3053 (=CH _st_), 1674 (C=O _st_) cm^−1^. HRMS (ESI-TOF) *m/z*: [M+H]^+^ calcd for C_25_H_25_N_2_O 369.1967, Found 369.1972.

3-Butyl-5-phenyl-1-(*p*-tolyl)-3-(*p*-tolylamino)-1,3-dihydro-2*H*-pyrrol-2-one (**25b**). The general procedure was followed, using 5-phenyl-1-(*p*-tolyl)-3-(*p*-tolylamino)-1,5-dihydro- 2*H*-pyrrol-2-one (0.354 g, 1 mmol) (**5a**) and a 1.6 M solution of *n*-butyllithium in hexanes (1.25 mL, 2 mmol), affording 300 mg (73%) of **25b** as an orange oil after chromatography (Hexanes/AcOEt 8:2). ^1^H NMR (400 MHz, CDCl_3_) δ 7.34–7.16 (m, 6H, 6 × CH_Ar_), 7.11 (d, ^3^*J*_HH_ = 8.3 Hz, 2H, 2 × CH_Ar_), 7.05–6.95 (m, 6H, 5 × CH_Ar_ + NH), 5.61 (s, 1H, =CH), 2.33 (s, 3H, CH_3_Tol), 2.29 (s, 3H, CH_3_Tol), 2.12 (m, 1H, CH_A_H_B_), 1.85 (m, 1H, CH_A_H_B_), 1.36–1.00 (m, 4H, 2 × CH_2_Bu), 0.76 (t, ^3^*J*_HH_ = 6.9 Hz, 3H, CH_3_Bu) ppm. ^13^C NMR {^1^H} (101 MHz, CDCl_3_) δ 175.0 (C_quat_), 166.6 (C=O), 148.6 (C_quat_), 136.2 (C_quat_), 135.2 (C_quat_), 133.0 (C_quat_), 131.9 (C_quat_), 129.9 (CH_Ar_), 129.5 (2 × CH_Ar_), 129.2 (2 × CH_Ar_), 128.6 (2 × CH_Ar_), 128.2 (2 × CH_Ar_), 127.0 (2 × CH_Ar_), 121.9 (2 × CH_Ar_), 95.5 (=CH), 95.0 (C_quat_), 38.0 (CH_2_Bu), 25.0 (CH_2_Bu), 22.6 (CH_2_Bu), 21.1 (CH_3_Tol), 21.0 (CH_3_Tol), 14.1 (CH_3_Bu) ppm. FTIR (neat) ν_max_: 3411 (NH _st_), 3053 (=CH _st_), 1703 (C=O _st_) cm^−1^. HRMS (ESI-TOF) *m/z*: [M+H]^+^ calcd for C_28_H_31_N_2_O 411.2436, Found 411.2442.

3-Butyl-1-(4-methoxyphenyl)-3-((4-methoxyphenyl)amino)-5-phenyl-1,3-dihydro-2*H*-pyrrol-2-one (**25c**). The general procedure was followed, using 1-(4-methoxyphenyl)-3-((4-methoxyphenyl)amino)-5-phenyl-1,5-dihydro-2*H*-pyrrol-2-one (386 mg, 1 mmol) (**5c**) and a 1.6 M solution of *n*-butyllithium in hexanes (1.25 mL, 2 mmol), affording 296 mg (67%) of **25c** as an orange oil after chromatography (Hexanes/AcOEt 7:3). ^1^H NMR (400 MHz, CDCl_3_) δ 7.34–7.20 (m, 6H, 5 × CH_Ar_ + NH), 7.06 (d, ^3^*J*_HH_ = 8.8 Hz, 2H, 2 × CH_Ar_), 7.05 (d, ^3^*J*_HH_ = 8.8 Hz, 2H, 2 × CH_Ar_), 6.87 (d, ^3^*J*_HH_ = 8.8 Hz, 2H, 2 × CH_Ar_), 6.74 (d, ^3^*J*_HH_ = 8.8 Hz, 2H, 2 × CH_Ar_), 5.63 (s, 1H, =CH), 3.79 (s, 3H, OCH_3_), 3.77 (s, 3H, OCH_3_), 2.08 (m, 1H, CH_A_H_B_), 1.82 (m, 1H, CH_A_H_B_), 1.31–1.07 (m, 4H, 2 × CH_2_Bu), 0.77 (t, ^3^*J*_HH_ = 6.8 Hz, 3H, CH_3_Bu) ppm. ^13^C NMR {^1^H} (101 MHz, CDCl_3_) δ 174.4 (C_quat_), 167.4 (C=O), 157.7 (C_quat_), 156.4 (C_quat_), 143.6 (C_quat_), 131.5 (C_quat_), 131.4 (C_quat_), 130.2 (CH_Ar_), 128.7 (2 × CH_Ar_), 128.7 (2 × CH_Ar_), 128.4 (2 × CH_Ar_), 123.1 (2 × CH_Ar_), 114.3 (2 × CH_Ar_), 113.9 (2 × CH_Ar_), 95.1 (C_quat_), 94.8 (=CH), 55.6 (OCH_3_), 55.5 (OCH_3_), 37.9 (CH_2_Bu), 25.1 (CH_2_Bu), 22.7 (CH_2_Bu), 14.1 (CH_3_Bu) ppm. FTIR (neat) ν_max_: 3391 (NH _st_), 3047 (=CH _st_), 1693 (C=O _st_) cm^−1^. HRMS (ESI-TOF) *m/z*: [M+H]^+^ calcd for C_28_H_31_N_2_O_3_ 443.2335, Found 443.2334.

1-(4-Bromophenyl)-3-((4-bromophenyl)amino)-3-butyl-5-phenyl-1,3-dihydro-2*H*-pyrrol-2-one (**25d**). The general procedure was followed, using 1-(4-bromophenyl)-3-((4-bromophenyl)amino)-5-phenyl-1,5-dihydro-2*H*-pyrrol-2-one (**5d**) (484 mg, 1 mmol) and a 1.6 M solution of *n*-butyllithium in hexanes (1.25 mL, 2 mmol), affording 411 mg (76%) of **25d** as yellow crystals after chromatography (Hexanes/AcOEt 85:15) followed by crystallization (Et_2_O /Pentane 1:3). M.p. (Et_2_O/Pentane) = 129–130 °C. ^1^H NMR (400 MHz, CDCl_3_) δ 7.42 (d, ^3^*J*_HH_ = 8.7 Hz, 2H, 2 × CH_Ar_), 7.34–7.10 (m, 10H, 9 × CH_Ar_ + NH), 6.94 (d, ^3^*J*_HH_ = 8.7 Hz, 2H, 2 × CH_Ar_), 5.55 (s, 1H, =CH), 1.37–1.03 (m, 5H, 5 × CHBu), 0.87 (m, 1H, CHBu), 0.74 (t, ^3^*J*_HH_ = 6.9 Hz, 3H, CH_3_Bu) ppm. ^13^C NMR {^1^H} (75 MHz, CDCl_3_) δ 175.3 (C_quat_), 167.5 (C=O), 150.6 (C_quat_), 138.8 (C_quat_), 132.1 (2 × CH_Ar_), 131.5 (C_quat_), 130.5 (CH_Ar_), 128.8 (2 × CH_Ar_), 128.7 (2 × CH_Ar_), 128.6 (2 × CH_Ar_), 127.3 (2 × CH_Ar_), 123.7 (2 × CH_Ar_), 116.8 (C_quat_), 108.2 (C_quat_), 95.5 (C_quat_), 94.9 (=CH), 37.9 (CH_2_Bu), 25.1 (CH_2_Bu), 22.7 (CH_2_Bu), 14.0 (CH_3_Bu) ppm. FTIR (neat) ν_max_: 3397 (NH _st_), 3056 (=CH _st_), 1692 (C=O _st_) cm^−1^. HRMS (ESI-TOF) *m/z*: [M+H]^+^-Br, calcd for C_26_H_26_BrN_2_O 463.1208, Found 463.1215.

3-Butyl-1-(3-chlorophenyl)-3-((3-chlorophenyl)amino)-5-phenyl-1,3-dihydro-2*H*-pyrrol-2-one (**25e**). The general procedure was followed, using 1-(3-chlorophenyl)-3-((3-chlorophenyl)amino)-5-phenyl-1,5-dihydro-2*H*-pyrrol-2-one (**5e**) (395 mg, 1 mmol) and a 1.6 M solution of *n*-butyllithium in hexanes (1.25 mL, 2 mmol), affording 338 mg (75%) of **25e** as yellow crystals after chromatography (Hexanes/AcOEt 85:15) followed by crystallization (Dichloromethane/Hexanes 1:3). M.p. (Dichloromethane/Hexanes) = 50–51 °C. ^1^H NMR (400 MHz, CDCl_3_) δ 7.39–7.21 (m, 7H, 7 × CH_Ar_), 7.13–7.09 (m, 2H, 2 × CH_Ar_), 7.08–7.02 (m, 2H, 2 × CH_Ar_), 6.94–6.87 (m, 2H, 2 × CH_Ar_), 5.57 (s, 1H, =CH), 2.05 (m, 1H, CH_A_H_B_), 1.86 (m, 1H, CH_A_H_B_), 1.32–1.02 (m, 4H, 2 × CH_2_Bu), 0.75 (t, ^3^*J*_HH_ = 7.0 Hz, 3H, CH_3_Bu) ppm. ^13^C NMR {^1^H} (101 MHz, CDCl_3_) δ 175.2 (C_quat_), 166.9 (C=O), 152.8 (C_quat_), 140.1 (C_quat_), 134.6 (C_quat_), 134.3 (C_quat_), 131.1 (C_quat_), 130.6 (CH_Ar_), 130.2 (CH_Ar_), 129.4 (CH_Ar_), 128.6 (2 × CH_Ar_), 128.5 (2 × CH_Ar_), 126.7 (CH_Ar_), 125.9 (CH_Ar_), 125.1 (CH_Ar_), 123.9 (CH_Ar_), 121.9 (CH_Ar_), 120.2 (CH_Ar_), 96.3 (C_quat_), 94.8 (=CH), 37.7 (CH_2_Bu), 24.9 (CH_2_Bu), 22.6 (CH_2_Bu), 13.9 (CH_3_Bu) ppm. FTIR (neat) ν_max_: 3391 (NH _st_), 3069 (=CH _st_), 1705 (C=O _st_) cm^−1^. HRMS (ESI-TOF) *m/z*: [M+H]^+^ calcd for C_26_H_25_Cl_2_N_2_O 451.1344, Found 451.1352.

### 3.2. Biology

#### 3.2.1. Materials

Reagents and solvents were used as purchased without further purification. All stock solutions of the investigated compounds were prepared by dissolving the powered materials in appropriate amounts of Dimethylsulfoxide (DMSO). The final concentration of DMSO never exceeded 5% (*v*/*v*) in reactions. The stock solution was stored at 5 °C until it was used.

#### 3.2.2. Cell Culture

Human epithelial lung carcinoma cells (A549) (ATCC^®^ CCL-185™, ATCC-Manassas, VA, United States) were grown in Kaighn’s Modification of Ham’s F-12 Medium (ATCC^®^ 30-2004™, ATCC-Manassas, VA, United States) and lung fibroblast cells (MRC5) (ATCC^®^ CCL-171™, ATCC-Manassas, VA, United States) were grown in Eagle’s Minimum Essential Medium (EMEM, ATCC^®^ 30-2003™, ATCC-Manassas, VA, United States). Epithelial ovary adenocarcinoma cells (SKOV3) (ATCC^®^ HTB-77™, ATCC-Manassas, VA, United States) were grown in McCoy’s 5A medium (ATCC^®^ 30-2007™, ATCC-Manassas, VA, United States). All of them were supplemented with 10% of fetal bovine serum (FBS) (Sigma-Aldrich, Spain) and with 1% of NORMOCIN solution (Thermo Fisher, Waltham, Massachusetts (MA), United States). Cells were incubated at 37 °C and 5% CO_2_ atmosphere, and were split every 3–4 days to maintain monolayer coverage. For cytotoxicity experiments, A549 and SKOV3 cells were seeded in 96-well plates at a density of 2.5–3 × 10^3^ cells per well and incubated overnight to achieve 70% of confluence at the time of exposition to the cytotoxic compound.

#### 3.2.3. Cytotoxicity Assays

Cells were exposed to different concentrations of the cytotoxic compounds and were incubated for 48 h. Then, 10 µL of cell counting kit-8 was added into each well for an additional two hours incubation at 37 °C. The absorbance of each well was determined by an Automatic Elisa Reader System (Thermo Scientific Multiskan FC Automatic Elisa Reader System, Thermo Scientific, Shangai, China) at 450 nm wavelength.

## 4. Conclusions

The Brönsted-acid-catalyzed multicomponent reaction of amines, aldehydes, and pyruvate derivatives is a straightforward method for the synthesis of 3-amino α,β-unsaturated γ-lactam derivatives. Selective functionalization at C-4 or C-5 of those substrates is feasible, making use of enamine chemistry or vinylogous nucleophilic additions, while the addition of organometallic species allows functionalization at C-3. Remarkably, the vinylogous aldol reaction with ethyl glyoxalate leads to the exclusive formation of the *anti*-adduct. In addition, chiral-functionalized γ-lactams are excellent precursors for diastereoselective synthetic transformations. The hydrogenation of the enamine moiety proceeds with high degree of *syn* diastereoselectivity, leading to the saturated substrates suitable for the further preparation of bicyclic compounds through intramolecular reactions. In addition, an example of a formal [3+3] annulation reaction is described with high *cis* stereocontrol. Moreover, some of the obtained γ-lactam derivatives showed in vitro cytotoxicity, inhibiting the growth of human tumor cells SKOV3 (human ovarian carcinoma) and A549 (carcinomic human alveolar basal epithelial cell) with selectivity toward MRC5 non-malignant lung fibroblasts. Although none of the substrates were proved to be superior to chemotherapeutic agent Doxorubicin, substrate **12b** showed a promising 3.6 μM IC_50_ value in A549 cell line, whereas the best result in SKOV3 cell line, 4.64 μM, was measured for compound **25e**.

## Data Availability

The data presented in this study are available in the Appendix A or on request from the corresponding author (^1^H, ^13^C, ^19^F, and ^31^P-NMR and HRMS spectra and cytotoxicity essays).

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
