# Peer review of "Exploring the Synthetic Potential of γ-Lactam Derivatives Obtained from a Multicomponent Reaction—Applications as Antiproliferative Agents"

_molecules, 2022, doi:10.3390/molecules27113624_

Round 1
Reviewer 1 Report
The work under review is devoted to the synthesis of γ-lactam derivatives as a result of multicomponent reactions, their selective functionalization at C-4 or C-5 carbon atoms, and the study of antitumor activity of the novel compounds against human ovarian carcinoma cells and alveolar basal epithelial cells. The article is written well and clearly, unfortunately, with some misprints. The structure of the new compounds has been sufficiently proven by 2D NMR and/or X-ray experiments where necessary. The article has some shortcomings.
1. The value of the radical E in Scheme 2 is inaccurate.
2. References 21e (p. 6), 11b (p. 9), 11a (p. 11), 59-0 (p. 12) in the text of the paper do not correspond to the List References.
The paper can be recommended for publication in the journal "Molecules" after a thorough editorial check with minor revision.
Author Response
We thank this referee for his/her very possitive comments. The following changes have been made
1. The value of the radical E in Scheme 2 is inaccurate.
Scheme 2 has been modified, according to both referee’s suggestions, and the structure of acetylenes has been added. The numbering in the whole document has been updated accordingly, since an additional number has been added to acetylenes.
2. References 21e (p. 6), 11b (p. 9), 11a (p. 11), 59-0 (p. 12) in the text of the paper do not correspond to the List References.
All the references have been checked and corrected.
Author Response
We thank this referee for his/her very possitive comments. The following changes have been made:
1. Scheme 1 Representation of “R2” has to be corrected.
Scheme 1 has been corrected
2. Scheme 2 Representation of “E” Must be Trance alkene not Alkyne.
Scheme 2 has been modified, according to both referee’s suggestions, and the structure of acetylenes has been added. The numbering in the whole document has been updated accordingly, since an additional number has been added to acetylenes.
3. The paragraph under the Scheme 2 should be rephrased as “However under the new set of reaction conditions such as stoichiometric amounts of a Lewis acid Yb(OTf)3 led to the formation of functionalized lactams 6g-h in excellent yields.
The sentence has been modified accordingly.
4. Scheme 8, starting material 21 is methyl ester whereas, product 22 is showing ethyl ester on generated six membered ring.
The structure of compound 22 (now 23) has been modified.
5. In my opinion the structure determination studies would be moved to supporting information under the spectral copies of corresponding compounds.
We agree with referee nr.2 that extensive structure determination studies are, normally, better placed in the supporting information. However, for this particular issue, named “Synthesis, Characterizations and Applications of Bioactive Heterocyclic Compounds”, we think it is interesting to keep those studies within the main text, thus covering the three aspects: synthesis, characterization and biological activity.
6. The introduction and conclusions of the manuscript are so lengthy can be written precisely.
The introduction and conclusions have been reduced. This implied a rearrangement of some of the references.